# Navigating Data Heterogeneity in Federated Learning: A Semi-Supervised Federated Object Detection

**Taehyeon Kim**[1]  **Eric Lin**[2]  **Junu Lee**[3]  **Christian Lau**[2]  **Vaikkunth Mugunthan**[2]

[1]KAIST  [2]DynamoFL  [3]The Wharton School

`potter32@kaist.ac.kr`

## Abstract

Federated Learning (FL) has emerged as a potent framework for training models across distributed data sources while maintaining data privacy. Nevertheless, it faces challenges with limited high-quality labels and non-IID client data, particularly in applications like autonomous driving. To address these hurdles, we navigate the uncharted waters of Semi-Supervised Federated Object Detection (SSFOD). We present a pioneering SSFOD framework, designed for scenarios where labeled data reside only at the server while clients possess unlabeled data. Notably, our method represents the inaugural implementation of SSFOD for clients with 0% labeled non-IID data, a stark contrast to previous studies that maintain some subset of labels at each client. We propose **FedSTO**, a two-stage strategy encompassing **S**elective **T**raining followed by **O**rthogonally enhanced full-parameter training, to effectively address data shift (e.g. weather conditions) between server and clients. Our contributions include selectively refining the backbone of the detector to avert overfitting, orthogonality regularization to boost representation divergence, and local EMA-driven pseudo label assignment to yield high-quality pseudo labels. Extensive validation on prominent autonomous driving datasets (BDD100K, Cityscapes, and SODA10M) attests to the efficacy of our approach, demonstrating state-of-the-art results. Remarkably, FedSTO, using just 20-30% of labels, performs nearly as well as fully-supervised centralized training methods.

## 1   Introduction

Federated Learning (FL) enables decentralized training across distributed data sources, preserving data privacy [27]. It has emerged in response to the need for privacy, security, and regulatory compliance such as GDPR [36] and CCPA [31]. FL trains models on local devices and shares only model updates, thereby improving privacy and efficiency. In a typical FL cycle, each client updates a shared model with local data, sends the updates to a server for parameter aggregation, and then updates its local model with the newly aggregated global model sent back by the server.

Despite the potential of FL, the assumption of fully labeled data restricts its practicality [12, 11]. In order to acquire high-quality labels, data is often transmitted from edge clients to a central server, thereby compromising the privacy assurances provided by FL. Limited labels at the edge necessitate the adoption of transfer learning, self-supervised learning, and semi-supervised learning (SSL) techniques. However, the separation of labeled and unlabeled data complicates the application of these techniques to FL, which can undermine the system's effectiveness. This issue is amplified in labels-at-server scenarios where only the server possesses labeled data, and clients hold only unlabeled data [5, 10, 43, 2, 18, 42]. In autonomous driving, a novel approach is required to bridge the knowledge gap between labeled and unlabeled data without the need for direct data exchange.

While Semi-Supervised Federated Learning (SSFL) has been explored for image classification tasks,[5, 10, 43, 2, 18, 42], these studies have faced the following challenges:

37th Conference on Neural Information Processing Systems (NeurIPS 2023).

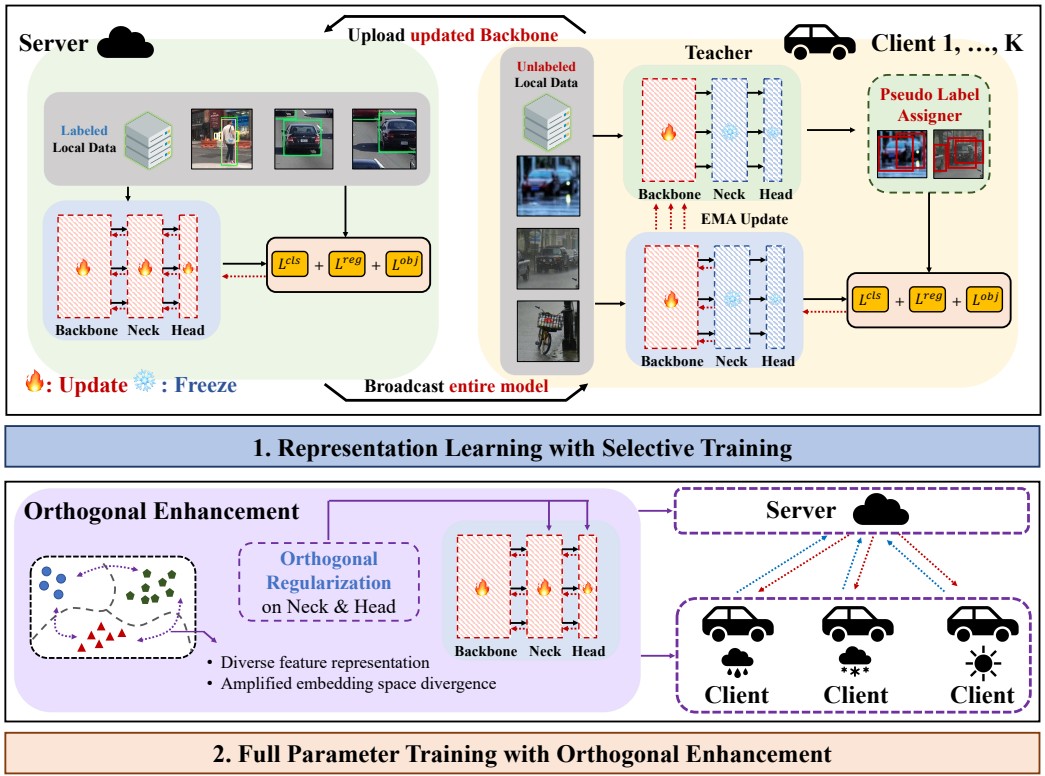

Figure 1: An overview of our FedSTO method within the SSFOD framework with key components: selective training, orthogonal enhancement, and local Exponential Moving Average (EMA)-driven pseudo label assignment, organized into two stages. Algorithm steps are numbered accordingly.

1. Limited scale and complexity of tasks with datasets such as CIFAR and ImageNet, while semi-supervised federated object detection (SSFOD) presents sizably greater difficulties.

2. Non-IID data shift from labeled to unlabeled data. Our investigation stands apart in tackling the most challenging FL situations where clients hold exclusively unlabeled data from a different distribution from labeled server data. This acknowledges inherent heterogeneity of real-world FL settings, such as diverse weather conditions across clients. For instance, one client's dataset may predominantly consist of images captured under cloudy conditions, while others may include images from overcast, rainy, snowy, etc. conditions.

To surmount these inadequately addressed challenges of SSFOD, we introduce FedSTO (Federated Selective Training followed by Orthogonally enhanced training), a two-stage training strategy tailored specifically for our SSFOD framework (Figure 1). Our key contributions include:

- **Selective Training and Orthogonal Enhancement:** FedSTO begins with selective training of the model's backbone while other components remain frozen, fostering more consistent representations and establishing a robust backbone. This promotes generalization across non-IID clients, even in the absence of local labels. The subsequent stage involves fine-tuning all parameters with orthogonal regularizations applied to the non-backbone part of the model. This enhancement step is designed to imbue the predictors with resilience against skewed representations induced by local data heterogeneity, thereby promoting representation divergence and robustness.

- **SSFL with a Personalized EMA-Driven Semi-Efficient Teacher:** To prevent deterioration of teacher pseudo labeling models for non-IID unlabeled clients, we showcase for the first time an SSFOD framework that applies an alternate training methodology [5], integrated with a Semi-Efficient Teacher [38], driven by a local Exponential Moving Average (EMA). Our empirical observations suggest that this personalized EMA-driven model provides superior quality pseudo labels for detection, contrary to the commonly used global model

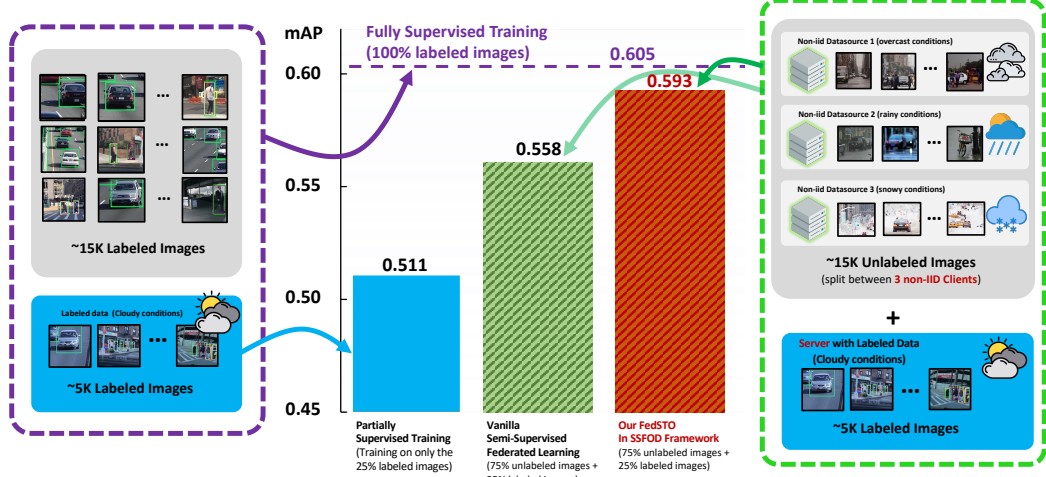

Figure 2: Performance comparison on BDD100K dataset [41]. "Partially Supervised Training" shows lower-bound performance using partial labels in a centralized setting. "Vanilla Semi-Supervised Federated Learning" and "Our FedSTO" demonstrate improved performance with non-IID federated data. FedSTO approaches the "Fully Supervised Training" upper-bound performance under full label use in a centralized setting. The x-axis shows the number of labeled examples, and the y-axis displays the mean average precision (mAP@0.5) on the test set.

> for pseudo labeling in related studies [5]. This approach further enhances the quality of the learning process, mitigating potential pitfalls of noisy pseudo labeling.

- **Performance Improvements:** FedSTO achieves 0.082 and 0.035 higher mAP@0.5 when compared to partially supervised and SSFL baselines respectively, nearly matching the fully supervised model's performance (0.012 gap) on BDD100K [41] with a mere 25% of labeled data. We demonstrate similar considerable improvements in model generalization (Figure 2) on rigorous benchmark and ablation experiments with 20k-120k datapoints from Cityscapes [4] and SODA10M [9], utilizing the YOLOv5 object detector [13].

Our above contributions present a pioneering approach for utilizing unlabeled data in FL to enhance non-IID detection performance, especially for dynamic objects—an aspect not yet considered in previous research. Despite the paucity of research on SSFOD, we contend that our methods and experiments offer a valuable benchmark for future investigations across diverse domains.

## 2 Related Works

### 2.1 Federated Learning (FL): Challenges and Advances

FL has gained significant attention in recent years as a privacy-preserving approach to harness the potential of distributed data [27, 20, 21, 15, 28, 33]. Despite the progress made in FL, most research has focused primarily on classification tasks, which may limit its applicability and generalizability to a broader range of real-world problems. Advanced FL techniques are essential to revolutionize various fields, including autonomous driving, healthcare, and finance, by enabling collaborative learning from distributed data sources [6, 30].

Addressing data heterogeneity is of paramount importance in FL, as clients frequently hold data with diverse distributions, which may impact the performance of the global model. To tackle this challenge, researchers have proposed various techniques to handle non-IID data, including adaptive aggregation algorithms and local fine-tuning of models [20, 33]. Personalization constitutes another vital aspect of FL, since clients may exhibit unique requirements or preferences not entirely captured by the global model [3, 19, 39]. Methods such as model distillation [23] and meta-learning [7] have been investigated to facilitate client-specific model adaptation and personalization. Finally, communication efficiency is crucial in FL, as exchanging model updates can be resource-intensive. To alleviate this issue, researchers have introduced strategies like quantization [34], sparsification [29], and the utilization of a supernet containing multiple subnetworks, with only the selected subnetwork transmitted to the server to reduce communication overhead while preserving model performance [17].

## 2.2 Semi-Supervised Object Detection (SSOD) with YOLO Object Detector

SSOD has been an active research area, focusing on improving the quality of pseudo labels to enhance overall detection performance [35, 40]. The evolution of SSL techniques in object detection primarily revolves around using pretrained architectures and applying strong data augmentation strategies to generate consistent and reliable pseudo labels.

Traditional single-stage detectors, such as the family of YOLO detectors, have faced notable challenges in leveraging SSL techniques, often underperforming compared to their two-stage counterparts (e.g., Faster RCNN). The limited efficacy of existing SSL methods for single-stage detectors has inspired researchers to develop innovative solutions to overcome these limitations [44]. Recently, a novel pipeline incorporating EMA of model weights has exhibited remarkable enhancements in the performance of single-stage detectors like YOLO detectors [38]. By utilizing the EMA model for pseudo labeling, researchers have adeptly addressed the inherent weaknesses in single-stage detectors, substantially elevating their performance in SSL contexts for object detection tasks.

## 2.3 Semi-Supervised Federated Learning (SSFL)

SSFL has emerged as a promising approach to address the challenge of limited labeled data in FL scenarios [5, 10, 43, 2, 18, 42]. SSFL aims to jointly use both labeled and unlabeled data owned by participants to improve FL. Two primary settings have been explored: Labels-at-Client and Labels-at-Server [10, 11]. In the Labels-at-Client scenario, clients possess labeled data, while the server only has access to unlabeled data. Conversely, in the Labels-at-Server scenario, the server holds labeled data, and clients have only unlabeled data. Despite the progress in SSFL, there remain limitations in the current research landscape. The majority of existing SSFL research predominantly focuses on image classification tasks, leaving other applications relatively unaddressed. In this study, we address these limitations by tackling the more realistic and challenging scenarios with edge clients having (1) no labels and (2) non-IID data (domain shift from the server labeled data), specifically in the context of object detection tasks.

## 3 Problem Statement

**SSFOD** We tackle a semi-supervised object detection task involving a labeled dataset $\mathcal{S} = \{\boldsymbol{x}_i^s, \boldsymbol{y}_i^s\}_{i=1}^{N_S}$ and an unlabeled dataset $\mathcal{U} = \{x_i^u\}_{i=1}^{N_U}$, focusing on scenarios where $N_S \ll N_U$. In our SSFOD setup, as illustrated in Figure 1, we assume $M$ clients each possessing an unsupervised dataset $x^{u,m}$. The server retains the labeled dataset $\{x^s, \boldsymbol{y}^s\}$ and a model parameterized by $W^s$. Each client model, parameterized by $W^{u,m}$, shares the object detection architecture denoted by $f : (\boldsymbol{x}, W) \mapsto f(\boldsymbol{x}, W)$. We assume that all models share the same object detection architecture, denoted by $f : (\boldsymbol{x}, W) \mapsto f(\boldsymbol{x}, W)$, which maps an input $\boldsymbol{x}$ and parameters $W$ to a set of bounding boxes and their corresponding class probabilities on the $K$-dimensional simplex (e.g., using the sigmoid function applied to model outputs unit-wise).

**Data Heterogeneity** Our study addresses non-IID data resulting from varying weather conditions such as cloudy, overcast, rainy, and snow, inspired by feature distribution skew or covariate shift [14]. We utilize three datasets, BDD100K [41], CityScapes [4], and SODA10M [9], each displaying class distribution heterogeneity and label density heterogeneity. Our aim is an SSFOD framework that can manage this heterogeneity, maintaining performance across diverse conditions and distributions. Data is considered IID when each client exhibits a balanced weather condition distribution.

**Evaluation** In our framework, we assess the performance of all detection tasks based on mean average precision (mAP@0.5), a standard metric in object detection literature that provides a comprehensive view of model performance across various object classes and sizes. Importantly, we evaluate the post-training performance of our method by assessing the personalized models of the server and client on their respective datasets. This approach ensures a fair and context-specific evaluation, reflecting the true performance of the personalized models in their intended environments.

**Baseline Training** Our work explores two principal baselines: "Centralized Training" and "Federated Learning". Depending on the degree of labeled data utilization, we categorize the training into "Partially Supervised" and "Fully Supervised". An ideal situation is one where a fully supervised model is trained in a centralized fashion, utilizing all labeled data. In contrast, a more challenging scenario involves a partially supervised model trained solely on the server's limited labeled data. Under our problem setup, we initially establish a baseline by performing partial supervision on the

Table 1: Performance under different weather conditions for non-IID and IID data splits with 1 server and 3 clients. The results are presented for centralized (Fully Supervised and Partially Supervised) and federated approaches with a pseudo label assigner (Global Model and Local EMA Model).

| Type | Method | Non-IID | | | | IID | | | |
|---|---|---|---|---|---|---|---|---|---|
| | | Cloudy | Overcast | Rainy | Snowy | Cloudy | Overcast | Rainy | Snowy |
| Centralized | Fully Supervised | 0.600 | 0.604 | 0.617 | 0.597 | 0.600 | 0.604 | 0.617 | 0.597 |
| | Partially Supervised | 0.540 | 0.545 | 0.484 | 0.474 | 0.528 | 0.545 | 0.533 | 0.510 |
| Federated | Global Model [5] | 0.555 | 0.560 | 0.497 | 0.488 | 0.540 | 0.551 | 0.576 | 0.542 |
| | Local EMA Model [38] | 0.560 | 0.566 | 0.553 | 0.553 | 0.572 | 0.588 | 0.593 | 0.610 |

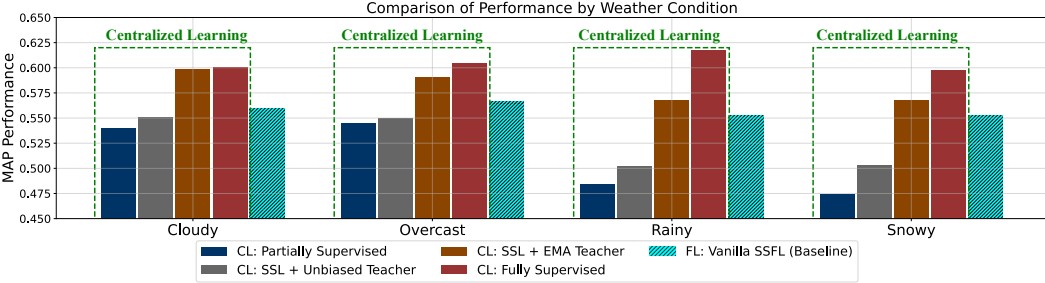

Figure 3: Performance of various methods on the BDD100K dataset [41], with the server containing labeled data for the "Cloudy" category and 3 clients having unlabeled data for "Rainy", "Snowy", and "Overcast" categories. Baseline SSFL (red hatched boxes) struggles in comparison to centralized learning (bars in green dotted boxes). "Fully Supervised" and "Partially Supervised" refer to training a centralized model with the complete labeled dataset and only the "Cloudy" labeled data, respectively.

server's limited labeled data, which serves as a pretraining step. Following this, each client conducts "Unsupervised learning with unlabeled data". Upon completion, clients transmit their model weights to the server. The server then aggregates these weights and fine-tunes the amalgamated model using its labeled data. The updated model is subsequently disseminated back to the clients, culminating one learning round. This cyclical process, known as alternate training in Diao et al. [5], continues effectively. It merges the strength of supervised and unsupervised learning to capitalize on unlabeled data while preventing model deterioration, thereby optimizing model performance.

**Personalized Pseudo Labeling for Unlabeled Clients**    A crucial obstacle in SSFOD lies in precise pseudo label assignment, as improper allotments can result in label inconsistencies, thus negatively impacting mutual learning performance. Building upon the foundation by Xu et al. [38] in centralized settings, we present the first extension of this approach to federated settings, leveraging a personalized Pseudo Label Assigner (PLA) equipped with local EMA models. This technique bifurcates pseudo labels into reliable and unreliable categories using high and low thresholds, thus ensuring a robust and precise learning mechanism in federated environments. In FL, the PLA can be applied to both global and local models. However, the global model may fall short in capturing unique features of local data, compromising pseudo label quality. As demonstrated in our evaluation (Table 1), locally updated EMA models outperform global models. While it is feasible to federate the local EMA model, it introduces certain trade-offs, such as increased communication costs and minor performance degradation compared to the local EMA model. Our SSFOD framework, therefore, incorporates a local PLA with a local EMA model, optimally balancing communication efficiency and model stability, ensuring an effective learning process for SSOD tasks in distributed environments.

**SSFOD with YOLO**    We utilize the YOLOv5 model, a single-stage object detector, in our evaluation. Existing literature shows a scarcity of research on SSFOD within FL like FedAvg [27], particularly for single-stage detectors like YOLO. Figure 3 compares various learning approaches in centralized and federated settings, denoted by green dotted and blue hatched boxes, respectively. We highlight non-IID scenarios with labeled (cloudy) and unlabeled data (overcast, rainy, snowy). In the CL scenario, fully supervised methods noticeably surpass partially supervised ones, and SSL approaches almost match the performance of fully supervised methods. However, baseline training for FL falls substantially short of these high standards, particularly with unlabeled data.

Table 2: Performance on the BDD dataset with 1 labeled server and 3 unlabeled clients as each element of our FedSTO approach within the SSFOD framework is added. It highlights how each added method contributes to the overall performance under both Non-IID and IID conditions.

| Method | Non-IID | | | | | IID | | | | |
|---|---|---|---|---|---|---|---|---|---|---|
| | Cloudy | Overcast | Rainy | Snowy | Total | Cloudy | Overcast | Rainy | Snowy | Total |
| Partially Supervised | 0.540 | 0.545 | 0.484 | 0.474 | 0.511 | 0.528 | 0.545 | 0.533 | 0.510 | 0.529 |
| + SSFL [5] with Local EMA Model | 0.560 | 0.566 | 0.553 | 0.553 | 0.558 | 0.572 | 0.588 | 0.593 | **0.610** | 0.591 |
| + Selective Training | 0.571 | 0.583 | 0.557 | 0.556 | 0.567 | 0.576 | 0.578 | 0.594 | 0.599 | 0.587 |
| + FPT with Orthogonal Enhancement [16] | **0.596** | **0.607** | **0.590** | **0.580** | **0.593** | **0.591** | **0.634** | **0.614** | 0.595 | **0.609** |

## 4 Main Method: FedSTO

To mitigate these inherent hurdles presented by FL, we introduce FedSTO, a method that unfolds in two stages, preceded by a warmup stage. The process commences with an emphasis on robust representation learning for pretraining (Subsection 4.1), followed by full parameter training (Subsection 4.2). The initial stage of pretraining integrates a warm-up period utilizing labeled data at the server, transitioning into selective training. This groundwork is fortified by the orthogonal enhancement implemented in the subsequent full parameter training phase.

### 4.1 Selective Training (ST)

Selective Training (ST) is designed to address the primary challenge of establishing a robust backbone for object detector in FL. The approach unfolds as follows:

1. **Labeled dataset training**: All model parameters are updated using a labeled dataset. This step ensures training commences on high quality labeled data, mitigating the potential destabilizing effect of noisy, unlabeled data, and heterogeneity from diverse weather conditions.

2. **Client-side training with unlabeled dataset**: The model, updated in the previous step, is dispatched to the clients. Each client trains the model on their local unlabeled dataset. However, during this phase, only the backbone part of the model is updated, leaving other components frozen. This selective updating procedure fosters more consistent representations by sharing the same non-backbone part (e.g., neck and head), and thus enhances its potential generalization capabilities by concentrating on feature extraction.

3. **Server-side aggregation**: The server aggregates the updated backbone parameters from clients, effectively synthesizing the learned information from diverse unlabeled datasets. The aggregated backbone is then utilized in the first step for further training, repeating the process until performance convergence.

By adhering to this procedure, ST effectively navigates the challenges inherent in the progression of FL while simultaneously accruing substantial benefits. Ensuring stability in semi-supervised object detection tasks is paramount. The exposure to heterogeneous unlabeled data, potentially characterized by noise or variable quality, can induce biases into the neck and head components of the model, thereby risking performance degradation by inadvertently generating low-quality or imprecise pseudo annotations. To mitigate this, ST employs a selective update strategy specifically targeting the backbone of the model, which is predominantly entrusted with the task of extracting salient features from the input data. By concentrating on the backbone during training, ST aids in the preservation of model stability and the enhancement of its generalization capabilities. Furthermore, in this stage, the communication cost between the server and clients is significantly reduced by uploading only the backbone part from clients to a server. Consequently, it significantly minimizes the deleterious impacts of heterogeneous unlabeled data on overall model performance (Table 2). While ST brings marginal improvements in IID conditions, it presents potent effects under Non-IID circumstances, emphasizing its efficacy in handling heterogeneous data distributions.

### 4.2 Full Parameter Training (FPT) with Orthogonal Enhancement

Inspired by the critical need for personalized models to exhibit robustness against feature distribution skewness—predominantly due to diverse weather conditions—we integrate orthogonality regularization presented by Kim et al. [16] which penalizes the symmetric version of the spectral restricted isometry property regularization $\sum_\theta \sigma(\theta^T\theta - I) + \sigma(\theta\theta^T - I)$ within the SSFOD framework where $\sigma(\cdot)$ calculates the spectral norm of the input matrix and $\theta$ is a weight matrix from non-backbone parts. This regularization is applied during both server and client training stages and targets non-backbone

**Algorithm 1:** FedSTO Algorithm within the SSFOD Framework

---

INPUT : server model parameterized by $W_s$, the number of rounds for each phase $T_1, T_2$, client models parameterized by $\{W_{u,1}, ..., W_{u,M}\}$, client backbone part parameterized by $\{B_{u,1}, ..., B_{u,M}\}$

1: $W_s \leftarrow \text{WARMUP}(x_s, y_s, W_s)$        `// Supervised training at server`

    /* Phase 1: Selective Training for Pretraining */

2: **for** $t \leftarrow 0, \dots, T_1 - 1$ **do**

3:    $S^t \leftarrow \text{SAMPLECLIENTS}$

4:    **for** each client $k \in S^t$ in parallel **do**

5:       $W_{u,k} \leftarrow \text{CLIENT-BACKBONEUPDATE}(x_{u,k}, B_{u,k})$       `// Client-Update`

6:    **end for**

7:    $W_s \leftarrow \sum_{k \in S^t} p_k W_{u,k}$              `// Aggregation`

8:    $W_s \leftarrow \text{SERVER-UPDATE}(x_s, y_s, W_s)$       `// Server-Update`

9: **end for**

    /* Phase 2: Full Parameter Training with Orthogonal Enhancement */

10: **for** $t \leftarrow 0, \dots, T_2 - 1$ **do**

11:    $S^t \leftarrow \text{SAMPLECLIENTS}$

12:    **for** each client $k \in S^t$ in parallel **do**

13:       $W_{u,k} \leftarrow \text{CLIENT-ORTHOGONALUPDATE}(x_{u,k}, W_{u,k})$

         `// Client-OrthogonalUpdate`

14:    **end for**

15:    $W_s \leftarrow \sum_{k \in S^t} p_k W_{u,k}$            `// Aggregation`

16:    $W_s \leftarrow \text{SERVER-ORTHOGONALUPDATE}(x_s, y_s, W_s)$    `// Server-OrthogonalUpdate`

17: **end for**

---

components of the architecture. Our approach promotes generation of diverse, non-redundant, and domain-invariant feature representations, thereby enhancing the model's robustness, reducing noise influence, and significantly augmenting its ability to handle unlabeled data across varied domains.

Incorporating orthogonality regularization into our framework substantially amplifies the divergence in the embedding space, enhancing the model's overall detection quality and the reliability of pseudo labels. Importantly, our strategy of embedding orthogonality into the non-backbone parts of the model, such as the neck and head, fosters a more balanced and comprehensive training process. This reduces the bias towards specific weather conditions and the heterogeneity of object categories, leading to improved performance as demonstrated in Table 2. Our approach draws upon successful techniques from fine-tuning [8, 26], and transfer learning, and is particularly inspired by meta-learning concepts,[37, 32]. In particular, the tendency of the non-backbone components of the model to develop biases prompts us to introduce an orthogonal property to this section. This measure helps counteract these biases, thereby further enhancing the model's robustness and adaptability when confronted with diverse, unlabeled data across multiple domains.

### 4.3 Main Algorithm: FedSTO

Algorithm 1 illustrates the overall procedure of FedSTO within the SSFOD framework. The server model, parameterized by $W_s$, is first trained in a supervised fashion during the warm-up phase (Line 1). The algorithm then transitions to Phase 1: Selective Training for Pretraining. This phase involves multiple training iterations (Line 3), where in each iteration, a subset of clients is sampled (Line 4). The backbone part of each client's model, $W_{u,k}$, is updated using their local unlabeled datasets (Line 6). The updated parameters are then aggregated at the server (Line 8), and the server model is updated using its labeled dataset (Line 9). In Phase 2: Full Parameter Training with Orthogonal Enhancement, the CLIENT-ORTHOGONALUPDATE and SERVER-ORTHOGONALUPDATE METHODS are employed (Lines 14 and 18), introducing orthogonality regularization to the training process. This second phase debiases the non-backbone parts of the model, ensuring they have a robust predictor across various weather conditions that effectively counterbalance the inherent data heterogeneity.

## 5 Experiment

### 5.1 Experimental Setup

#### 5.1.1 Datasets

**BDD100K [41]** We utilize the BDD100K dataset, which consists of 100,000 driving videos recorded across diverse U.S. locations and under various weather conditions, to evaluate our method. Each

Table 3: Comparison of FedSTO within the SSFOD framework against the Baselines, SSL, SSFL methods with 1 server and 3 clients on BDD100K dataset [41]. FedSTO exhibits improvements under various weather conditions on both IID and Non IID cases, and performs close to the centralized fully supervised case. † denotes the SSFL with the local EMA model as a pseudo label generator.

| Type | Algorithm | Method | Non-IID | | | | | IID | | | | |
|---|---|---|---|---|---|---|---|---|---|---|---|---|
| | | | Cloudy | Overcast | Rainy | Snowy | Total | Cloudy | Overcast | Rainy | Snowy | Total |
| Centralized | SL | Fully Supervised | 0.600 | 0.604 | 0.617 | 0.597 | 0.605 | 0.600 | 0.604 | 0.617 | 0.597 | 0.605 |
| | | Partially Supervised | 0.540 | 0.545 | 0.484 | 0.474 | 0.511 | 0.528 | 0.545 | 0.533 | 0.510 | 0.529 |
| | SSL | Unbiased Teacher [25] | 0.551 | 0.550 | 0.502 | 0.503 | 0.527 | 0.546 | 0.557 | 0.541 | 0.533 | 0.544 |
| | | EMA Teacher [38] | 0.598 | 0.59 | 0.568 | 0.568 | 0.581 | 0.586 | 0.570 | 0.571 | 0.573 | 0.575 |
| Federated | SFL | Fully Supervised | 0.627 | 0.614 | 0.607 | 0.585 | 0.608 | 0.635 | 0.612 | 0.608 | 0.595 | 0.613 |
| | SSFL† | FedAvg [27] | 0.560 | 0.566 | 0.553 | 0.553 | 0.558 | 0.572 | 0.588 | 0.593 | **0.610** | 0.591 |
| | | FedDyn [1] | 0.508 | 0.569 | 0.541 | 0.522 | 0.535 | 0.355 | 0.414 | 0.420 | 0.397 | 0.400 |
| | | FedOpt [33] | 0.561 | 0.572 | 0.565 | 0.566 | 0.566 | 0.591 | 0.587 | 0.588 | 0.577 | 0.586 |
| | | FedPAC [39] | 0.514 | 0.532 | 0.496 | 0.489 | 0.508 | 0.510 | 0.549 | 0.547 | 0.554 | 0.540 |
| | | **FedSTO** | **0.596** | **0.607** | **0.590** | **0.580** | **0.593** | **0.591** | **0.634** | **0.614** | 0.595 | **0.609** |

video, approximately 40 seconds in duration, is recorded at 720p and 30 fps, with GPS/IMU data available for driving trajectories. For our experiments, we specifically select 20,000 data points, distributed across four distinct weather conditions—cloudy, rainy, overcast, and snowy. In this study, we primarily focus on five object categories: person, car, bus, truck, and traffic sign. The dataset is partitioned into clients based on these weather conditions, simulating data-heterogeneous clients. This experimental setup enables us to investigate the influence of data heterogeneity on our framework and to evaluate its robustness under realistic conditions.

**Cityscape [4]** We conduct additional experiments using the Cityscapes dataset, which consists of urban street scenes from 50 different cities. Given that this dataset does not provide precise weather information for each annotation, we distribute the data to clients in a uniformly random manner. For our studies, we employ the package, encompassing fine annotations for 3,475 images in the training and validation sets, and dummy annotations for the test set with 1,525 images. We also include the other package, providing an additional 19,998 8-bit images for training.

**SODA10M [9]** To evaluate our approach under diverse conditions, we employ the SODA10M dataset, which features varied geographies, weather conditions, and object categories. In an IID setup, 20,000 labeled data points are uniformly distributed among one server and three clients. For a more realistic setup, the 20,000 labeled data points are kept on the server while 100,000 unlabeled data points are distributed across the clients. This arrangement enables performance evaluation under distinct weather conditions—clear, overcast, and rainy—showcasing resilience and robustness.

### 5.1.2 Training Details

We conduct our experiments in an environment with one server and multiple clients, depending on the experiment. Both the server and the clients operate on a single local epoch per round. Our training regimen spans 300 rounds: 50 rounds of warm-up, 100 rounds of pretraining ($T_1$), and 150 rounds of orthogonal enhancement ($T_2$). We use the YOLOv5 Large model architecture with Mosaic, left-right flip, large scale jittering, graying, Gaussian blur, cutout, and color space conversion augmentations. A constant learning rate of 0.01 was maintained. Binary sigmoid functions determined objectiveness and class probability with a balance ratio of 0.3 for class, 0.7 for object, and an anchor threshold of 4.0. The ignore threshold ranged from 0.1 to 0.6, with an Non-Maximum Suppression (NMS) confidence threshold of 0.1 and an IoU threshold of 0.65. We incorporate an exponential moving average (EMA) rate of 0.999 for stable model parameter representation.

### 5.2 Results

Table 3 illustrates the efficacy of our proposed SSFOD method against various baselines and state-of-the-art approaches on the BDD100K dataset. FedSTO significantly outperforms other techniques under different weather conditions and data distribution scenarios, i.e., IID and Non-IID. In the CL scenarios, the fully supervised approach yields the highest performance, with SSL methods, such as EMA Teacher [38], demonstrating competitive results. However, the real challenge lies in federated settings, where data privacy and distribution shift become critical considerations. In the SSFOD framework, our FedSTO method consistently surpasses other SSFL techniques. Notably, it achieves superior results even in challenging Non-IID settings, demonstrating its robustness to data distribution shifts. Similar trends hold when increasing the number of clients as shown in the appendix. In IID

Table 4: Performance under random distributed cases of Cityscapes [4]. FedSTO exhibits improvements under various object categories, and significantly outperforms the performance for unlabeled clients. † denotes the SSFL with the local EMA model as a local pseudo label generator.

| Type | Algorithm | Method | Labeled | | | | | Unlabeled | | | | |
|---|---|---|---|---|---|---|---|---|---|---|---|---|
| | | | Categories | | | | | | | | | |
| | | | Person | Car | Bus | Truck | Traffic Sign | Person | Car | Bus | Truck | Traffic Sign |
| Centralized | SL | Fully Supervised | 0.569 | 0.778 | 0.530 | 0.307 | 0.500 | 0.560 | 0.788 | 0.571 | 0.283 | 0.510 |
| | | Partially Supervised | 0.380 | 0.683 | 0.193 | 0.302 | 0.246 | 0.358 | 0.648 | 0.343 | 0.138 | 0.255 |
| | SSL | Unbiased Teacher [25] | 0.391 | 0.695 | 0.225 | 0.320 | 0.297 | 0.410 | 0.689 | 0.373 | 0.129 | 0.354 |
| | | EMA Teacher [38] | 0.475 | 0.711 | 0.354 | 0.347 | 0.379 | 0.460 | 0.727 | 0.436 | 0.144 | 0.378 |
| Federated | SFL | Fully Supervised | 0.498 | 0.715 | 0.357 | 0.289 | 0.410 | 0.492 | 0.714 | 0.451 | 0.251 | 0.425 |
| | SSFL† | FedAvg [27] | 0.450 | 0.697 | 0.310 | **0.304** | 0.356 | 0.482 | 0.725 | 0.425 | **0.247** | 0.397 |
| | | FedBN [22] | 0.488 | 0.709 | 0.325 | 0.285 | 0.411 | 0.375 | 0.618 | 0.046 | 0.031 | 0.286 |
| | | **FedSTO** | **0.504** | **0.720** | **0.342** | 0.261 | **0.415** | **0.487** | **0.740** | **0.460** | 0.181 | **0.437** |

conditions, our method continues to excel, achieving results close to fully supervised centralized approach. These findings highlight the strength of our FedSTO method in leveraging the benefits of FL while mitigating its challenges. The robust performance of our approach across various weather conditions and data distributions underscores its potential for real-world deployment.

When examining the performance on the Cityscapes dataset under uniformly random distributed conditions, the superiority of FedSTO within the SSFOD framework also remains apparent, as shown in Table 4. Compared to other methods, FedSTO consistently demonstrates improved generalization across most object categories, both for labeled and unlabeled data. Intriguingly, the performance of FedSTO surpasses even that of SSL in CL environments.

**Evaluation with mAP@0.75** The mAP@0.75 results on the BDD dataset highlight the efficacy of the FedSTO approach (Table 5). In Non-IID settings, while the Fully Supervised centralized method achieve an average mAP of 0.357, FedSTO recorded 0.338, exhibiting comparable performance. However, under IID conditions, FedSTO registers an mAP@0.75 of 0.357, closely matching the SFL result of 0.359. These results indicate that FedSTO offers competitive object detection capabilities, even with stricter IoU thresholds.

Table 5: mAP@0.75 on the BDD dataset with 1 labeled server and 3 unlabeled clients.

| Type | Algorithm | Method | Non-IID | | | | | IID | | | | |
|---|---|---|---|---|---|---|---|---|---|---|---|---|
| | | | Cloudy | Overcast | Rainy | Snowy | Total | Cloudy | Overcast | Rainy | Snowy | Total |
| Centralized | SL | Fully Supervised | 0.351 | 0.352 | 0.368 | 0.356 | 0.357 | 0.351 | 0.352 | 0.368 | 0.356 | 0.357 |
| | | Partially Supervised | 0.281 | 0.295 | 0.261 | 0.303 | 0.285 | 0.281 | 0.295 | 0.261 | 0.303 | 0.285 |
| Federated | SFL | Fully Supervised | 0.384 | 0.366 | 0.357 | 0.317 | 0.356 | 0.377 | 0.356 | 0.352 | 0.349 | 0.359 |
| | SSFL | FedSTO W/O ST | 0.321 | 0.303 | 0.265 | 0.267 | 0.289 | 0.298 | 0.321 | 0.324 | 0.302 | 0.311 |
| | | FedSTO | 0.347 | 0.351 | 0.341 | 0.312 | 0.338 | 0.343 | 0.375 | 0.361 | 0.350 | 0.357 |

**Results on Real World Dataset, SODA10m [9]** Figure 4a illustrates the performance of our method and other baselines on the SODA10m dataset, where labeled data is synthetically divided in an IID manner across one server and three clients. Our method demonstrates near-parity with the fully supervised approach, evidencing its efficacy. Figure 4b represents the averaged performance across varying weather conditions on the SODA10m dataset. Here, all 20k labeled data resides on the server, and 100k unlabeled data points from SODA10m are distributed across three clients. Despite these variations in conditions, our method consistently outperforms other baselines, confirming its robustness and applicability in diverse environments.

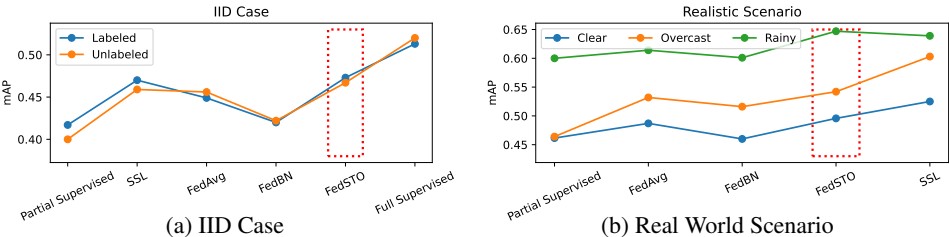

(a) IID Case    (b) Real World Scenario

Figure 4: (a) Performance of various methods on the SODA10m dataset in an IID setting, (b) Average performance across different weather conditions using unlabeled data from the SODA10m dataset.

**Varying Number of Clients** In a non-IID BDD dataset configuration with 1 server and 20 clients, our proposal advances beyond competing methods, scoring 0.455 and 0.458 on labeled and unlabeled data, respectively. This outcome showcases our method's aptitude for tackling intricate real-world circumstances.

Table 6: Performance on a non-IID case of the BDD100k dataset with 1 server and 20 clients.

| Type | Centralized | Federated-SSFL | | |
|---|---|---|---|---|
| Method | Partially Supervised | FedAvg | ST | FedSTO |
| Labeled | 0.3768 | 0.405 | 0.405 | **0.455** |
| Unlabeled | 0.3524 | 0.4311 | 0.4322 | **0.458** |

**Varying Sampling Ratio** Table 7 demonstrates the impact of different client sampling ratios on the FedSTO performance using the BDD100k dataset. Notably, even at a lower sampling ratio of 0.1, FedSTO yields commendable results, especially in the unlabeled set for categories like 'Car' (0.738) and 'Bus' (0.573). This underscores that a reduced client sampling can still lead to significant performance improvements, emphasizing the efficiency and adaptability of the FL approach.

Table 7: Performance (mAP@0.5) under Non-IID scenarios of BDD100k dataset with 1 server and 100 clients according to the changes of client sampling ratio for implementing FedSTO. The term 'Server Only' aligns with the notion of 'partially supervised' in CL settings.

| Method | Labeled | | | | | Unlabeled | | | | |
|---|---|---|---|---|---|---|---|---|---|---|
| | Categories | | | | | | | | | |
| | Person | Car | Bus | Truck | Traffic Sign | Person | Car | Bus | Truck | Traffic Sign |
| Server Only (i.e., client sampling ratio 0.0) | 0.378 | 0.710 | 0.141 | 0.425 | 0.490 | 0.337 | 0.707 | 0.160 | 0.338 | 0.491 |
| FedSTO with client sampling ratio 0.1 | 0.393 | 0.714 | 0.442 | 0.510 | 0.540 | 0.487 | **0.738** | **0.573** | **0.589** | **0.617** |
| FedSTO with client sampling ratio 0.2 | **0.458** | **0.747** | **0.476** | **0.521** | **0.571** | 0.440 | 0.731 | 0.378 | 0.525 | 0.573 |
| FedSTO with client sampling ratio 0.5 | 0.444 | 0.745 | 0.437 | 0.502 | 0.550 | **0.489** | 0.730 | 0.438 | 0.512 | 0.538 |

**Efficiency on Network Bandwidth** Table 8 highlights the communication costs over 350 rounds of training involving 100 clients with a 0.5 client sampling ratio per round. By removing the neck component of the Yolov5L model, its size is reduced from 181.7MB to 107.13MB. This reduction significantly benefits FedSTO in Phase 1, leading to overall bandwidth savings. When comparing with traditional SSFL methods such as FedAvg and FedProx,[20], FedSTO utilizes only **2,166.23 GB** - a substantial **20.52%** reduction in network bandwidth.

Table 8: Communication costs over 350 rounds of training with 100 clients when the client sampling ratio is 0.5 per each round. The total Yolov5L size is 181.7MB while the model without the neck part is 107.13MB. Additionally, the model size without BN layers (FedBN [22]) is 181.24 MB. Here, 'Reduction' expresses how much communication cost is reduced compared to using vanilla SSFL (FedAvg and FedProx [20]).

| Method | Warm-up (50 rounds) | Phase 1 (150 rounds) | Phase 2 (150 rounds) | Total | Reduction |
|---|---|---|---|---|---|
| FedAvg FedProx | 0 | 100 * 0.50 * 150 * 181.7 = 1,362.75 GB | 100 * 0.50 * 150 * 181.7 = 1,362.75 GB | 2,725.50 GB | - |
| FedBN | 0 | 100 * 0.50 * 150 * 181.24 = 1359.30 GB | 100 * 0.50 * 150 * 181.24 = 1359.30 GB | 2,718.60 GB | 0.25 % |
| FedSTO | 0 | 100 * 0.50 * 150 * 107.13 = 803.48 GB | 100 * 0.50 * 150 * 181.7 = 1,362.75 GB | **2,166.23 GB** | **20.52 %** |

# 6 Conclusion

This paper introduces a novel Semi-Supervised Federated Object Detection (SSFOD) framework, featuring a distinctive two-stage training strategy known as FedSTO. Designed to address the challenges of heterogeneous unlabeled data in federated learning, FedSTO employs selective training and orthogonality regularization with personalized pseudo labeling. These mechanisms facilitate robust and diverse feature learning, thereby enhancing object detection performance across multiple weather conditions and data distributions. Empirical results provide compelling evidence of the superiority of FedSTO over established federated and semi-supervised learning methodologies. Notably, despite operating with the challenging constraint where non-IID clients have no labels, FedSTO successfully counteracts domain shift and achieves performance that is comparable to fully supervised centralized models. This accomplishment constitutes significant strides toward realizing more efficient and privacy-preserving learning in realistic FL settings. As we venture ahead, we aim to concentrate our research efforts on refining FedSTO and exploring additional strategies for leveraging unlabeled data with various domains and model architectures. We anticipate the work presented in this paper will stimulate continued progress in this rapidly evolving field.

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

# A    Overview of Appendix

In this supplementary material, we present additional details that are not included in the main paper due to the space limit.

# B    Ethics Statement

In the pursuit of progress within the domain of Federated Learning (FL), we propose a novel method, FedSTO within the SSFOD. This innovation warrants an examination of its ethical ramifications, particularly in regards to privacy, fairness, environmental impact, and potential for misuse.

**Privacy and Data Security**    Analogous to established FL methodologies, SSFOD is architected to preserve the privacy of client data. By conducting computations locally and transmitting solely model updates, the risks associated with raw data transmission are mitigated. Nonetheless, the potential threat of adversarial actions, such as model inversion attacks, underscores the imperative for ongoing efforts to bolster the robustness of FL methods against such vulnerabilities.

**Fairness**    The deployment of FedSTO into SSFOD can either amplify or alleviate existing fairness concerns within FL, contingent on the particulars of its application. Should SSFOD predominantly utilize data stemming from specific demographic cohorts, the model predictions risk acquiring an inadvertent bias. In contrast, SSFOD's capacity to manage large-scale, real-world scenarios may engender a more diverse data inclusion, thereby fostering a more equitable model.

**Environmental Impact**    Similar to its FL counterparts, our approach diminishes the requirement for centralized data storage and computation, thereby potentially reducing associated carbon footprints. However, the energy expenditures of localized computation and communication for model updates necessitate judicious management to uphold environmental sustainability.

**Potential Misuses**    Although our approach is designed to enhance the robustness of FL methodologies in large-scale, real-world settings, the potential for misuse remains. For instance, malevolent entities could exploit FedSTO with SSFOD framework to deliberately introduce bias or disinformation into models. Consequently, the implementation of protective measures against such misuse is of paramount importance.

In summary, while our method represents a significant contribution to the FL field, its potential ethical implications mandate thoughtful application. We strongly endorse ongoing discourse and scrutiny to ensure that its deployment is in alignment with the principles of privacy, fairness, and social responsibility.

# C    Limitations

While our SSFOD method offers significant advancements in Semi-Supervised Federated Learning for Object Detection, it is important to recognize its accompanying limitations:

1. **Performance with Highly Imbalanced Data** Our SSFOD method exhibits robustness across a range of data heterogeneity. However, its performance in scenarios involving severe data imbalance across clients necessitates additional exploration. Imbalance here refers to disparities in data class distribution across clients. Instances of extreme skewness could lead to biased learning outcomes, with the model excelling in classes with abundant samples and faltering in those with fewer. Although SSFOD incorporates mechanisms to offset effects of data heterogeneity, it is not explicitly designed to manage extreme data imbalance.

2. **Computational Overhead** Despite its effectiveness in bolstering model robustness in large-scale, real-world scenarios, SSFOD introduces additional computational overhead. This is primarily due to the algorithmic complexities and computational requirements intrinsic to our method, which might be a constraint for resource-limited devices often participating in federated learning. This could potentially limit the scalability and applicability of our method in real-world FL scenarios. Therefore, improving the computational efficiency of SSFOD without compromising its efficacy is an important direction for future research.

3. **Sensitivity to Varying Weather Conditions** SSFOD has been designed to tackle the challenge of varying weather conditions in autonomous driving. However, in real-world scenarios, there are other types of environmental changes that can equally affect the learning process. For instance, varying lighting conditions or changes in road surfaces might influence the input data. Since our SSFOD method primarily focuses on weather conditions, it might not exhibit the same level of efficiency when dealing with other environmental factors. Future iterations of SSFOD could explore these areas to provide a more comprehensive solution to environmental handling in FL.

## D Future Directions

Despite the aforementioned limitations, our SSFOD method lays the groundwork for several promising future research directions:

1. **Handling Other Environmental Factors:** Future work could extend the SSFOD method to handle other environmental factors efficiently. This would make it a more comprehensive solution for real-world federated learning scenarios where different environmental factors coexist.

2. **Adaptation for Imbalanced Data:** Investigating and enhancing the performance of SSFOD with highly imbalanced data distribution would be a valuable future direction. Techniques like adaptive resampling or cost-sensitive learning could be integrated with our method to tackle this challenge.

3. **Optimization of Computational Efficiency:** Future research could focus on optimizing the computational efficiency of the SSFOD method. Reducing the computational overhead without compromising the robustness in large-scale, real-world scenarios would make our method more practical for real-world FL scenarios.

4. **Robustness Against Adversarial Attacks:** As the FL domain evolves, adversarial attacks pose an increasing threat to model robustness. Future work could explore how to bolster the SSFOD method (and FL methods, in general) to ensure robustness against adversarial attacks.

By addressing these limitations and exploring these future directions, we can continuously refine and evolve the SSFOD method to better serve the ever-growing demands of federated learning.

## E Detailed Data Heterogeneity in Federated Object Detection

Heterogeneity, a prevalent attribute in object detection datasets, often arises from three crucial aspects:

- **Weather-induced feature distribution skew:** Within outdoor scenarios like autonomous driving, weather variations significantly impact the visual representation of objects. Differing conditions such as sunny, rainy, foggy, or snowy can alter an object's appearance, causing the weather-induced feature distribution skew. Moreover, sensor diversity, including RGB cameras and infrared sensors, contributes to this skew as they respond uniquely to various weather conditions. This complex scenario creates a challenging task for an object detection system that must generalize across diverse conditions.

- **Class distribution heterogeneity:** This heterogeneity refers to the uneven representation of various classes within a dataset. In many cases, certain classes are far more prevalent than others. For instance, in an autonomous driving scenario, 'cars' or 'pedestrians' may be much more frequent than 'bicycles' or 'motorcycles.' This imbalance can cause learning algorithms to develop a bias towards more common classes. Moreover, in a federated learning scenario, the class distribution may vary among clients; a rural area client might capture more 'animal' instances compared to an urban area client.

- **Label density heterogeneity:** This form of heterogeneity pertains to the variation in the quantity of annotated objects per image. An image of a crowded scene may contain far more objects than a sparser image. This variability can influence the performance of detection models, particularly those that rely on a fixed number of anchors or proposals per image.

Furthermore, it can also impact the training process as images with more objects provide more "training signal" per image than those with fewer objects. In a federated learning context, certain clients might possess more densely labeled data than others, which could affect the learning process.

While our current work primarily addresses the issue of weather-induced feature distribution skew, the other forms of heterogeneity, i.e., class distribution heterogeneity and label density heterogeneity, also require careful consideration. They present unique challenges within the federated learning environment and significantly influence model performance. It is crucial to extend our methodologies to account for these factors, fostering more robust and versatile object detection systems capable of handling the intricate realities of real-world scenarios. Consequently, future work should aim at developing comprehensive solutions that cater to all these facets of data heterogeneity in the context of federated learning.

## F Exploring the Impact of Orthogonal Enhancement under Data Heterogeneity in SSFOD

### F.1 Theoretical Analysis

This section presents a straightforward bounding strategy for the loss in the context of data heterogeneity. Our primary focus is to ascertain the Mean Squared Error (MSE) of a two-layered neural network, encompassing both 'head' and 'backbone'. Analyzing this model's behavior under a slightly perturbed data distribution is intended to offer insights into worst-case loss scenarios when orthogonal weights are employed.

We consider a multivariate distribution, denoted by $\mathbb{P}$, with $\boldsymbol{x} \in \mathbb{R}^d$ and $\boldsymbol{y} \in \mathbb{R}^m$ being drawn from this distribution. We have $n$ i.i.d. samples from $\mathbb{P}$, constituting our dataset $\mathcal{D} = (\boldsymbol{x}_i, \boldsymbol{y}_i) \colon (\boldsymbol{x}_i, \boldsymbol{y}_i) \sim \mathbb{P}, 1 \leq i \leq n$. For the sake of this analysis, our model is expressed as $f(\boldsymbol{x}) = W^\top B \boldsymbol{x}$, where $B \in \mathbb{R}^{k \times d}$ and $W \in \mathbb{R}^{k \times m}$ are the backbone and head weight matrices, respectively. The MSE is given by $\mathbb{E}_{\boldsymbol{x}, \boldsymbol{y}} \|\boldsymbol{y} - f(\boldsymbol{x})\|_2^2$, representing the expectation of the $\ell^2$-norm of the difference vector between the ground truth and the prediction.

Under the presumption that $f$ has been finely optimized on the training sample distribution (using the dataset $\mathcal{D}$) to yield an MSE of value $L$, we next examine a perturbed data distribution, denoted by $\mathbb{P}'$, characterized as follows:

$$(\boldsymbol{x}', \boldsymbol{y}') \sim \mathbb{P}' \iff (\boldsymbol{x}', \boldsymbol{y}') \sim (\boldsymbol{x} + \boldsymbol{\epsilon}, \boldsymbol{y}) \tag{1}$$

Here, $(\boldsymbol{x}, \boldsymbol{y}) \sim \mathbb{P}$ and $\boldsymbol{\epsilon} \sim \mathcal{N}_d(0, \Sigma)$ is Gaussian noise, characterized by covariance matrix $\Sigma$, independent of $(\boldsymbol{x}, \boldsymbol{y})$. The MSE loss under $\mathbb{P}'$, represented by $L'$, can be bound as:

$$
\begin{aligned}
\mathbb{E}_{\boldsymbol{x}', \boldsymbol{y}'} \|\boldsymbol{y}' - W^\top B \boldsymbol{x}'\|_2^2 &= \mathbb{E}_{\boldsymbol{x}, \boldsymbol{y}, \boldsymbol{\epsilon}} \|\boldsymbol{y} - W^\top B(\boldsymbol{x} + \boldsymbol{\epsilon})\|_2^2 \\
&= \mathbb{E}_{\boldsymbol{x}, \boldsymbol{y}, \boldsymbol{\epsilon}} \|\boldsymbol{y} - W^\top B \boldsymbol{x} + W^\top B \boldsymbol{\epsilon}\|_2^2 \\
&= \mathbb{E}_{\boldsymbol{x}, \boldsymbol{y}} \|\boldsymbol{y} - W^\top B \boldsymbol{x}\|_2^2 + \mathbb{E}_{\boldsymbol{\epsilon}} \|W^\top B \boldsymbol{\epsilon}\|_2^2 \\
&\quad + \mathbb{E}_{\boldsymbol{x}, \boldsymbol{y}, \boldsymbol{\epsilon}} [(\boldsymbol{y} - W^\top B \boldsymbol{x})^\top W^\top B \boldsymbol{\epsilon}] \\
&\quad + \mathbb{E}_{\boldsymbol{x}, \boldsymbol{y}, \boldsymbol{\epsilon}} [\boldsymbol{\epsilon}^\top B^\top W(\boldsymbol{y} - W^\top B \boldsymbol{x})] \\
&= L + \mathbb{E} \|W^\top B \boldsymbol{\epsilon}\|_2^2.
\end{aligned}
\tag{2}
$$

This calculation introduces a nonnegative penalty of $\mathbb{E} \|W^\top B \boldsymbol{\epsilon}\|_2^2$ to the out-of-sample MSE.

The derived penalty opens the possibility for further analysis, particularly when considering structural assumptions on the head, backbone, or noise vector. The Gaussian assumption for the noise vector is quite enlightening, given that the squared $\ell^2$-norm is tantamount to the quadratic form $\boldsymbol{\epsilon}^\top B^\top W W^\top B \boldsymbol{\epsilon}$. By incorporating a recognized principle about the expectation of a Gaussian vector's quadratic form, we obtain:

$$\mathbb{E} \|W^\top B \boldsymbol{\epsilon}\|_2^2 = \mathbb{E}[\boldsymbol{\epsilon}^\top B^\top W W^\top B \boldsymbol{\epsilon}] = \operatorname{tr}(B^\top W W^\top B \Sigma). \tag{3}$$

Assuming $WW^\top = I$, an identity matrix, the trace simplifies to $\mathrm{tr}(B^\top B\Sigma)$. In the event the model $f$ has undergone orthogonality regularization, resulting in a semi-orthogonal head $W$ (precisely, $WW^\top = I$), we obtain a concise formulation for the penalty that hinges solely on the backbone and the fundamental noise parameter.

## F.2 Insights from Theoretical Analysis

The insights gained from our theoretical analysis significantly enrich our understanding of the FedSTO approach and how it can navigate challenges specific to FL.

One major challenge in FL is data heterogeneity – the data across different clients (or devices) can vary significantly. By bounding the MSE for our model under slightly shifted distributions, we learn how our model's performance could change in the presence of such data heterogeneity. This is akin to having a 'sensitivity' measure for our model, showing us how 'sensitive' the model is to changes in data distributions. The better we understand this sensitivity, the more effectively we can tailor our model to handle the challenges of federated learning.

The analysis also highlights the significance of having orthogonal weights in our model. The investigation reveals that when the head of our model is semi-orthogonal, the penalty on the out-of-sample MSE simplifies to depend only on the backbone and the noise term, effectively isolating the effect of client-specific data perturbations. This insight is of particular importance because it provides a clear and quantifiable understanding of the role and benefit of orthogonal weights in controlling model sensitivity to data heterogeneity.

Furthermore, our analysis provides valuable directions for future research in FL. The relationship discovered between the orthogonality of weights and the bound on MSE opens up new avenues for investigation. It could, for instance, prompt more in-depth research into regularization techniques that aim to achieve orthogonality, improving model stability across diverse data distributions.

In simpler terms, our theoretical analysis is much like a 'roadmap', helping us understand how our model reacts to changes in data distributions, the role of orthogonal weights in this context, and where we could focus future research efforts to improve our approach. This roadmap makes the journey of applying FedSTO to federated learning more navigable, ultimately leading to more effective and robust models.

# G   Detailed Experimental Settings

**YOLOv5 Architecture**   The YOLOv5 object detection model is based on the YOLO algorithm, which divides an image into a grid system and predicts objects within each grid. YOLOv5 uses a convolutional neural network (CNN) to extract features from the image and then uses these features to predict the location of the bounding boxes (x,y,height,width), the scores, and the objects classes. The YOLOv5 architecture consists of three main parts: the backbone, the neck, and the head. The backbone is responsible for extracting features from the image. The neck combines the features from the backbone and passes them to the head. The head then interprets the combined features to predict the class of an image. Here is a more detailed description of each part of the YOLOv5 architecture:

- Backbone: The backbone of YOLOv5 is based on the CSPDarknet53 architecture. The CSPDarknet53 architecture is a modified version of the Darknet53 architecture that uses a cross stage partial connection (CSP) module to improve the performance of the network. The CSP module allows the network to learn more complex features by sharing information across different layers.

- Neck: The neck of YOLOv5 is a simple convolutional layer that combines the features from the backbone.

- Head: The head of YOLOv5 is composed of three convolutional layers that predict the location of the bounding boxes, the scores, and the objects classes. The bounding boxes are predicted using a regression model, while the scores and classes are predicted using a classification model.

YOLOv5 has been shown to be effective for object detection, achieving good results on a variety of object detection benchmarks. It is also fast, making it a good choice for real-time object detection applications.

**Annotations**   The YOLO object detection algorithm divides an image into a grid system and predicts objects within each grid. The annotations for YOLO are stored in a text file, with each line representing an object in the image. The format of each line is as follows:

[object_id], [x_center], [y_center], [width], [height], [score]

where object_id is the ID of the object, x_center is the x-coordinate of the center of the object's bounding box, y_center is the y-coordinate of the center of the object's bounding box, width is the width of the object's bounding box, height is the height of the object's bounding box, and score is the confidence score of the object detection.

**Loss Functions**   YOLOv5 is an object detection model that is trained using a loss function that combines three terms: a class loss, a box loss, and a confidence loss. The class loss is a cross-entropy loss that measures the difference between the predicted class probabilities and the ground truth class labels. The box loss is a smooth L1 loss that measures the distance between the predicted bounding boxes and the ground truth bounding boxes. The confidence loss is a binary cross-entropy loss that measures the difference between the predicted confidence scores and the ground truth binary labels indicating whether or not an object is present in the image. The overall loss function is minimized using stochastic gradient descent. Here is a more detailed explanation of each loss term:

- Class loss $\mathcal{L}_{cls}$: The cross-entropy loss is a measure of the difference between two probability distributions. In the case of YOLOv5, the two probability distributions are the predicted class probabilities and the ground truth class labels. The cross-entropy loss is minimized when the predicted class probabilities are identical to the ground truth class labels.

- Box loss $\mathcal{L}_{obj}$: The smooth L1 loss is a measure of the distance between two sets of numbers. In the case of YOLOv5, the two sets of numbers are the predicted bounding boxes and the ground truth bounding boxes. The smooth L1 loss is minimized when the predicted bounding boxes are identical to the ground truth bounding boxes.

- Confidence loss $\mathcal{L}_{conf}$: The binary cross-entropy loss is a measure of the difference between two binary distributions. In the case of YOLOv5, the two binary distributions are the predicted confidence scores and the ground truth binary labels indicating whether or not an object is present in the image. The binary cross-entropy loss is minimized when the predicted confidence scores are identical to the ground truth binary labels.

## H   Discussions on Semi-Supervised Object Detection

We implement a similar strategy to the pseudo label assigner of the semi-efficient teacher [38]. We aim to provide a comprehensive description of our training method and pseudo label assignment approach in this section.

### H.1   Adaptive Loss

The adaptive loss function in our SSFOD framework is comprised of two key parts: a supervised loss ($L_s$), calculated from labeled data, and an unsupervised loss ($L_u$), generated from unlabelled instances.

The supervised loss, $L_s$, adheres to conventional practices used in object detection tasks. This incorporates the amalgamation of cross-entropy loss, responsible for classification, and CIoU (Complete Intersection over Union) loss, accounting for bounding box regression:

$$L_s = \sum_{h,w} \left( CE(X_{cls}^{(h,w)}, Y_{cls}^{(h,w)}) + CIoU(X_{reg}^{(h,w)}, Y_{reg}^{(h,w)}) + CE(X_{obj}^{(h,w)}, Y_{obj}^{h,w}) \right), \quad (4)$$

where $CE$ is the cross-entropy loss function, $X^{(h,w)}$ signifies the model's output, and $Y^{(h,w)}$ corresponds to the sampled results of pseudo label assigner (i.e., local EMA updated model).

The unsupervised loss component, $L_u$, is an addition that leverages the pseudo labels generated by the local EMA updated model. The objective of $L_u$ is to guide the model to exploit beneficial information embedded within unlabelled data. It is computed as:

$$L_u = L_{cls}^u + L_{reg}^u + L_{obj}^u, \tag{5}$$

where $L_{cls}^u$, $L_{reg}^u$, and $L_{obj}^u$ denote the losses associated with classification, bounding box regression, and objectness score, respectively. Each of these losses uses pseudo labels and is precisely tailored to foster reliable and efficient learning from unlabeled data.

- Classification Loss, $L_{cls}^u$, hones the model's class-specific accuracy. It is calculated only for pseudo labels whose scores are above a predefined high threshold $\tau_2$, using a cross-entropy loss function. The divergence is gauged between the model's predicted classification scores and the class scores from the pseudo labels provided by the Pseudo Label Assigner.

- Regression Loss, $L_{reg}^u$, scrutinizes the precision of the bounding box predictions. This loss applies to pseudo labels with scores above $\tau_2$ or objectness scores surpassing a value typically set at 0.99. A CIoU loss function is employed to measure the discrepancy between the predicted bounding box regressions and those associated with the pseudo labels.

- Objectness Loss, $L_{obj}^u$, evaluates the model's objectness prediction capability, essentially the model's confidence in a given bounding box containing an object of interest. All pseudo labels contribute to this loss, though the computation varies depending on the pseudo label score. For scores below $\tau_1$, the loss is computed against zero. For those above $\tau_2$, the loss compares against the objectness of the pseudo label. Scores between $\tau_1$ and $\tau_2$ result in the loss calculated against the soft objectness score of the pseudo labels.

This strategic amalgamation of losses forms the underpinning of the unsupervised loss function, fostering efficient and reliable learning from unlabeled data. By harnessing the potential information embedded within pseudo labels, our model aims to significantly enhance semi-supervised object detection in an FL setup.

In a centralized setting, the supervised and unsupervised losses would typically be trained jointly. However, in our federated scenario, where the server possesses only labeled data and the client holds exclusively unlabeled data, we adopt an alternate training approach. The server focuses on optimizing the supervised loss $L_s$ with its labeled data, while the client concurrently refines the unsupervised loss $L_u$ using its unlabeled data. This methodical division of labor allows us to leverage the distinctive characteristics of both types of data and facilitates efficient learning in a federated environment.

### H.2    Local EMA Model for the Pseudo Labeler

In SSFOD framework, we employ a Local Exponential Moving Average (EMA) model for generating pseudo labels, enabling the judicious utilization of unlabeled data at each client's disposal. The concept of the EMA model hinges on an infinite impulse response filter that assigns weightages decreasing exponentially, offering a balanced consideration of both historical and immediate data points for reliable predictions.

Each client maintains a local EMA model in our method. The weights of this local EMA model are a weighted average of its own weights and the weights of the global model. This relationship can be described mathematically as follows:

$$W_{\text{EMA}_{clientk}}^{(t)} = \alpha W_{\text{EMA}_{clientk}}^{(t-1)} + (1-\alpha)W_{\text{client k}}^{(t)} \tag{6}$$

In Eq. (6), $W_{\text{EMA}_{clientk}}^{(t)}$ symbolizes the weights of client k's local EMA model at round $t$, and $W_{\text{client k}}^{(t)}$ represents the weights of the client k's model at round $t$. $\alpha$ is the EMA decay rate dictating the pace of depreciation of the client's weights' influence. Typically, the decay rate resides in the 0.9 to 0.999 range, subject to the specific application.

Table 9: Performance on the BDD dataset with 1 labeled server and 3 unlabeled clients. It highlights how each added method contributes to the overall performance under both Non-IID and IID conditions.

| Method | Non-IID | | | | | IID | | | | |
|---|---|---|---|---|---|---|---|---|---|---|
| | Cloudy | Overcast | Rainy | Snowy | Total | Cloudy | Overcast | Rainy | Snowy | Total |
| Vanilla | 0.560 | 0.566 | 0.553 | 0.553 | 0.558 | 0.572 | 0.588 | 0.593 | **0.610** | 0.591 |
| Freezing Only Head | 0.531 | **0.603** | **0.567** | **0.565** | **0.567** | 0.558 | **0.613** | **0.614** | 0.593 | **0.595** |
| Freezing Neck & Head | **0.571** | 0.583 | 0.557 | 0.556 | **0.567** | **0.576** | 0.578 | 0.594 | 0.599 | 0.587 |

In pursuit of a consistent basis across all clients, the weights of the Local EMA model are reinitialized with the global model's weights post each server broadcast. This particularity of our framework ensures a harmonious coexistence of global model consistency and local data awareness.

The system enables the local EMA model of each client to gradually tune to the unique local data characteristics, by integrating updates from the client's unlabeled data. As a consequence, we achieve a potent blend of personalization and model performance in a FL environment, all the while alleviating communication overheads. The design acknowledges the distinct data distributions that each client may harbor, allowing the model to adapt accordingly, thereby augmenting the learning process's efficacy and efficiency.

Following the weight updates, the local EMA model functions as the pseudo labeler for each client. This approach ensures the generation of stable and reliable pseudo labels, despite the limited interactions with the global model. In the Federated Learning scenario, where communication costs carry substantial significance, this feature is crucial.

Our local EMA strategy presents several benefits. It guarantees independent pseudo label generation at each client's end without necessitating frequent server updates. In addition, it enhances the learning process's robustness, as the local EMA model remains largely unaffected by the client model's noisy gradient updates. As a result, the system produces more trustworthy pseudo labels, ultimately culminating in improved performance.

# I Discussions on Selective Training

In our SSFOD framework, Selective Training plays a pivotal role in achieving superior performance and enhanced computational efficiency. Traditionally, in selective training, all components beyond the backbone, typically referred to as Non-Backbone parts, are frozen to reduce computational complexity and improve training efficiency. However, It may not be necessary to freeze all of the non-backbone parts.

In our preliminary investigations, we discover that freezing only the head, a component of the Non-Backbone parts, can yield similar performance levels as freezing the entire Non-Backbone. This observation opens the door to potentially more efficient and flexible models in the context of selective training, as it suggests that selective freezing could be just as effective as complete freezing (Table 9).

Nonetheless, in consideration of the inherent nature of FL, we ultimately decide to continue freezing both the neck and head components. We reach this decision primarily due to the communication cost considerations, which carry considerable weight in a federated learning setting. Despite the promising results observed when only freezing the head, freezing both the neck and head does not result in any noticeable performance decrement, while it does markedly reduce the communication cost.

The implications of this choice are particularly relevant in the context of FL environments, where minimizing communication costs is paramount for practical deployment. We posit that this finding may inform future strategies for selective training, suggesting that selective freezing of specific Non-Backbone parts can effectively balance performance and efficiency in FL environments.

Table 10: Performance under random distributed cases of Cityscapes [4]. FedSTO exhibits improvements under various object categories, and significantly outperforms the performance for unlabeled clients.

| Method | Labeled | | | | | Unlabeled | | | | |
|---|---|---|---|---|---|---|---|---|---|---|
| | Categories | | | | | | | | | |
| | Person | Car | Bus | Truck | Traffic Sign | Person | Car | Bus | Truck | Traffic Sign |
| ORN from scratch | 0.470 | 0.694 | **0.449** | 0.074 | 0.393 | 0.437 | 0.720 | 0.507 | 0.100 | 0.378 |
| Freezing only Head + FPT with ORN | **0.504** | 0.697 | 0.353 | 0.239 | 0.411 | 0.486 | **0.740** | 0.420 | 0.078 | 0.415 |
| Freezing Neck & Head + FPT with ORN | **0.504** | **0.720** | 0.342 | **0.261** | **0.415** | **0.487** | **0.740** | **0.460** | **0.181** | **0.437** |

## J  Further Discussions on Full Parameter Training (FPT) with Orthogonal Enhancement

As part of our ongoing evaluation of our SSFOD framework, we conduct comprehensive experiments involving different training strategies, in particular, Full Parameter Training (FPT) with Orthogonal Regularization (ORN). Through these experiments, we compare three distinct approaches:

1. ORN employed right from scratch without any pre-training or frozen components.

2. A two-stage process where the model head is frozen during the pre-training phase, followed by fine-tuning across the entire model with ORN.

3. A more nuanced strategy where both the neck and head components of the model are initially frozen, subsequently transitioning to an ORN-driven FPT.

Our empirical results decisively point to the superiority of the third approach (Table 10). Despite the initial freezing of the neck and head components, it yields performance that surpassed the other two strategies. This counter-intuitive finding underscores the effectiveness of the careful balancing of selective parameter freezing and ORN application.

The mere marginal performance difference in mAP between the second and third approaches suggests room for further investigation. An exploration of the specific conditions under which one might outperform the other could shed light on new avenues for model improvement. This represents an exciting direction for future research.

However, for the scope of the present study, we opt for the third strategy of freezing both the neck and head components simultaneously. This decision is not only influenced by the slightly higher performance but also by the added advantage of reduced communication costs. This approach resonates with the primary objective of our research - to realize high-performing, efficient, and cost-effective FL frameworks for SSOD tasks.

## K  Examination of Loss Functions within SSFOD Framework

An integral part of our study on the SSFOD framework involves an examination of the influence of various loss functions on model performance. One of the notorious challenges in object detection tasks is dealing with class imbalance. To counter this, we decided to evaluate the effectiveness of Focal Loss, which is widely used for its capability to handle such imbalances, in comparison with the conventional Cross-Entropy (CE) Loss, which we refer to as vanilla Loss in the paper.

Focal Loss [24] has been designed to tackle class imbalance by reducing the contribution of easy instances, thus allowing the model to concentrate on challenging, misclassified samples. Despite its recognized efficacy in a variety of settings, we observed unexpected performance under our specific framework.

Contrary to our initial assumptions, our empirical findings suggest that the Vanilla Loss proved to be a more potent safeguard against the class imbalance issue within our SSFOD framework (Table 11). We compare the performance by changing the loss functions in the baseline SSFOD training. The underlying reasons for this surprising result are potentially multifaceted and certainly merit additional exploration. However, these observations underscore the importance of empirical testing in choosing the appropriate loss functions tailored for specific frameworks and tasks.

Table 11: Performance of CE loss and focal loss on the BDD dataset with 1 labeled server and 3 unlabeled clients. It highlights how each added method contributes to the overall performance under both Non-IID and IID conditions.

| Method | Non-IID | | | | | IID | | | | |
|---|---|---|---|---|---|---|---|---|---|---|
| | Cloudy | Overcast | Rainy | Snowy | Total | Cloudy | Overcast | Rainy | Snowy | Total |
| CE loss | **0.560** | **0.566** | **0.553** | **0.553** | **0.558** | **0.572** | **0.588** | **0.593** | **0.610** | **0.591** |
| Focal loss | 0.364 | 0.462 | 0.438 | 0.446 | 0.428 | 0.371 | 0.464 | 0.469 | 0.482 | 0.447 |

## L  Implementation

To provide a comprehensive overview of our study, we provide the details of our implementation, including the reproduction of various existing FL methods such as FedDyn [1], FedBN [22], FedOpt [33], and FedPAC [39].

- FedDyn [1]: For the implementation of FedDyn, we adhere strictly to the training methodology as stipulated in the original paper.

- FedBN [22]: In implementing the FedBN method, we make a conscientious choice to leave out the statistics and bias of the batch normalization (BN) layer from the aggregation step.

- FedOpt [33]: FedOpt implementation aligns with Vanilla SSFL, albeit with a slight modification in the server model's optimization procedure. The key differentiation lies in the utilization of a pseudo learning rate of 0.9 during server model optimization, which was chosen through empirical evaluations to maximize performance. Thus, while preserving the core attributes of the FedOpt framework, we adapt it to our specific task requirements.

- FedPAC [39]: The execution of the FedPAC framework, primarily architected for classification tasks, necessitates thoughtful adaptation to suit our object detection scenario. This adaptation involves the appropriation of the YOLOv5 head, used as a stand-in classifier, which allows us to transpose our specific needs onto the FedPAC blueprint. This process not only honors the training techniques prescribed by FedPAC but also satisfies our model's unique requirements. As an integral part of our approach, the local EMA model, a cornerstone of our broader learning strategy, is employed as a pseudo labeler in this scheme. Furthermore, our model's backbone representation is equated with the output from YOLO's backbone. This bespoke modification offers a perspective to exploit the central principles of FedPAC while tailoring them to suit the idiosyncrasies of our model.

## M Experimental Results with 100 Clients

In the setting of an extensive FL environment with a network of 100 clients and 0.1 client sampling ratio, we embark on an empirical investigation to ascertain the robustness and scalability of the proposed method. This scenario closely mirrors real-world cross-device situations, inherently characterized by widespread client distribution. Intriguingly, each client in our experiment is associated with data corresponding to a single weather condition. Notwithstanding this, our method exhibits remarkable resilience and efficacy. As an illustration, the model outperforms a baseline that is trained solely on the server's labeled data, as demonstrated in Table 12. This superior performance underscores the model's capability in effectively leveraging the heterogeneity intrinsic to distributed data, thereby enhancing the overall performance. These empirical findings provide compelling evidence of the adaptability and effectiveness of our approach within large-scale FL contexts. With its ability to maintain high performance coupled with computational efficiency, our method exhibits promising potential in managing data heterogeneity across extensive federated networks.

Table 12: Performance under Non-IID scenarios of BDD100k dataset with 1 server and 100 clients. FedSTO exhibits improvements under various object categories, and significantly outperforms the performance for unlabeled clients. The term 'Server Only Scenario' aligns with the notion of 'partially supervised' in CL settings.

| Method | Labeled | | | | | Unlabeled | | | | |
|---|---|---|---|---|---|---|---|---|---|---|
| | Categories | | | | | | | | | |
| | Person | Car | Bus | Truck | Traffic Sign | Person | Car | Bus | Truck | Traffic Sign |
| Server Only Scenario | 0.378 | 0.710 | 0.141 | 0.425 | 0.490 | 0.337 | 0.707 | 0.160 | 0.338 | 0.491 |
| FedSTO within SSFOD framework | **0.393** | **0.714** | **0.442** | **0.510** | **0.540** | **0.487** | **0.738** | **0.573** | **0.589** | **0.617** |

