# OpenReview forum: "Navigating Data Heterogeneity in Federated Learning: A Semi-Supervised Federated Object Detection"
_NeurIPS.cc/2023/Conference — NeurIPS 2023 poster_

### Official Review · Reviewer_1XGa · 2023-07-05

**Soundness:** 2 fair
**Presentation:** 2 fair
**Contribution:** 3 good
**Rating:** 5
**Confidence:** 4

**Summary:**

To solve the challenges with limited high-quality labels and non-IID client data in federated learning, the authors present a pioneering SSFOD framework, designed for scenarios where labeled data reside only at the server while clients possess unlabeled data. Meanwhile, they propose the FedSTO, which consists of selective training, orthogonal enhancement, and personalized EMA-driven semi-efficient teacher. Finally, FedSTO achieves 0.082 and 0.035 higher mAP@0.5 when compared to partially supervised and SSFL baselines respectively.

**Strengths:**

1. Good motivation. On the one hand, selective training can address the primary challenge of establishing a robust backbone for object detectors in FL. Specifically, it fosters more consistent representations by sharing the same non-backbone part. On the other hand, orthogonal enhancement reduces the bias towards specific weather conditions and the heterogeneity of object categories, leading to improved performance.
2. Many experiments have also verified the effectiveness of the proposed FedSTO.

**Weaknesses:**

1. The authors do not detailed introduce Personalized Pseudo Labeling for Unlabeled Clients, which is not easy to understand.
2. I don't understand what Fig.3 wants to express, why there will be Fully supervised in Unlabeled overcast, rainy, and snowy.
3. In Fig.1, I don't know how the orthogonal enhancement works on the neck and head. Besides, I can not understand how to generate pseudo labels and assign them to the client models.

**Questions:**

1. What do T1 and T2 in Algorithm 1 represent?

**Limitations:**

Yes, the authors have described limitations.

---

> ### Author Rebuttal · Authors · 2023-08-10
>
> ### W1. No explanations for personalized pseudo-labeling for unlabeled clients
> - Thank you for your question. We apologize if the presentation of the “Personalized Pseudo Labeling for Unlabeled Clients” was not clear in the main body of the paper due to space constraints. We have provided a comprehensive explanation in **Appendix H.2** which offers more details on this aspect.
> - In our SSFOD framework, we adopt a local Exponential Moving Average (EMA) model for generating pseudo labels, which facilitates efficient utilization of each client's unlabeled data. The EMA model operates on an infinite impulse response filter that assigns decreasing exponential weights. This arrangement ensures a balanced consideration of both historical and immediate data points, aiding in making reliable predictions.
> - Each client maintains a local EMA model whose weights are a weighted average of the client's own weights and the weights of the global model. This correlation is depicted in **Eq. (6) in the Appendix**. After each server broadcast, the weights of the Local EMA model are reinitialized with the global model’s weights, ensuring global model consistency along with local data awareness.
> - The client's local EMA model gradually tunes to the unique characteristics of its local data, integrating updates from the client's unlabeled data. Consequently, we create a balance of personalization and model performance in a FL environment while alleviating communication overheads.
> - Following the weight updates, the local EMA model functions as the pseudo labeler for each client. This method ensures the generation of stable and reliable pseudo labels, despite the limited interactions with the global model. The local EMA model guarantees independent pseudo label generation at each client’s end without requiring frequent server updates. It also enhances the learning process’s robustness, as the local EMA model remains largely unaffected by the client model’s noisy gradient updates.
> - We believe this explanation provides a better understanding of our approach. We hope to edit the main text for clarity and to add more details if we get extra pages for the camera-ready version of the paper. **For now, please refer to Appendix H.2 for a detailed understanding.**
>
> ### W2. The purpose of Fig 3
> - Thank you for your question about Figure 3. We apologize if it caused any confusion. In this figure, “Fully supervised” refers to the model trained with labeled data across all weather conditions—Cloudy, Overcast, Rainy, and Snowy. “Partially supervised” signifies the model trained only on labeled data from the “Cloudy” condition, while “Vanilla SSFL” implies the model trained on labeled data from 'Cloudy' and unlabeled data from “Overcast”, “Rainy”, and “Snowy” conditions. The performance of these models was then evaluated across all weather conditions, regardless of their training status.
> - We understand that the labeling of the figure might be a bit confusing, and we appreciate your feedback. We will work on clarifying the labeling in the figure to avoid any such confusion in the future.
>
> ### W3. Overview of the method
> - Thank you for your inquiry regarding Fig.1 and the processes involved.
> - The Orthogonal Enhancement functions by imposing an orthogonality regularization on the neck and head of the model (line 217). This process penalizes the kernel weight matrices to be orthogonal, being robust towards the feature bias (theoretically supported in the Appendix Section F).
> - Regarding the generation of pseudo labels, this is accomplished using a pseudo labeler. As outlined in Appendix Section G, the pseudo labeler generates predictions which then undergo a non-maximum suppression step to form pseudo annotations. These pseudo annotations are subsequently used in conjunction with the loss function described in the same section of the appendix, guiding the learning of the client models.
> - Due to space constraints in the main body of the paper, we have detailed these implementation specifics in the appendix. We appreciate your understanding and welcome any further queries you might have.
>
> ### Q1. What are T1 and T2 in Algorithm 1?
> - T1 and T2 in Algorithm 1 represent distinct phases in the learning process. Specifically, T1 refers to the number of pretraining rounds, which means “Representation Learning with Selective Training”. This phase aids in establishing a decent starting point for subsequent learning. On the other hand, T2 denotes the number of rounds dedicated to Orthogonal Enhancement. This phase is primarily designed to increase the diversity of feature learning by discouraging the kernel weight matrices from becoming orthogonal.
> - We will further clarify it by explicitly mentioning T1 and T2.

---

> > ### Author Response · Authors · 2023-08-16
> >
> > Dear Reviewer,
> >
> > - In response to the feedback, we've conducted rigorous additional experiments to enhance the depth and robustness of our work. This includes:
> > 1. Real-world experiments using 100 clients through 100 virtual machines.
> > 2. An extensive analysis of communication costs.
> > 3. Further ablation studies for a more comprehensive understanding.
> > - Additionally, we'd like to highlight that several points brought up during the review process have been addressed in our appendix. We've emphasized these in our updated responses for your convenience.
> > - Given the tight timeline, with the discussion phase concluding on Aug 21st at 1pm EDT, we kindly request you to review our responses. We believe our detailed responses provide clarity on the concerns raised. Your feedback is pivotal to the quality of our work, and we earnestly await your thoughts, especially since we have less than 6 days remaining.
> >
> > Thank you for your time and understanding.

---

> > > ### Comment · Reviewer_1XGa · 2023-08-16
> > >
> > > Thank you for your response, which addresses most of my concerns. I would like to increase my score to a 5 and suggest the final version to incorporate those clarifications stated in the rebuttal.

---

> > > > ### Author Response · Authors · 2023-08-16
> > > >
> > > > Dear Reviewer,
> > > >
> > > > Thank you for acknowledging the clarifications we provided in our response. We're pleased to hear that we addressed most of your concerns. We are committed to incorporating those clarifications into the final version of the paper, ensuring it reflects the insights gained from this review process.
> > > >
> > > > Additionally, if you're considering adjusting your score, you should be able to do so in the current review platform. We greatly appreciate your time and constructive feedback throughout this process.

---

### Official Review · Reviewer_AM8c · 2023-07-06

**Soundness:** 3 good
**Presentation:** 3 good
**Contribution:** 2 fair
**Rating:** 6
**Confidence:** 4

**Summary:**

This paper introduces a novel framework called Semi-Supervised Federated Object Detection (SSFOD) to tackle the problem of object detection in a federated learning setting. In this framework, the server possesses labeled data, while the clients hold unlabeled data from different distributions. The proposed approach consists of two stages: selective training and orthogonal enhancement. In the selective training stage, the focus is on updating only the backbone parameters on the clients to establish a robust backbone for the object detector. This selective approach helps improve the model's generalization capabilities across different distributions. The orthogonal enhancement stage follows, where all parameters are fine-tuned with orthogonal regularization. This regularization promotes representation divergence and robustness, further enhancing the model's performance. The paper also introduces a personalized pseudo label assigner based on a local exponential moving average (EMA) model. This assigner generates high-quality pseudo labels for object detection tasks, facilitating the training process in the semi-supervised setting. To evaluate the proposed SSFOD framework, this paper conducts experiments on three datasets: BDD100K, Cityscapes, and SODA10M. The results demonstrate that the proposed method achieves state-of-the-art performance when compared to existing approaches in both semi-supervised and federated learning domains.

**Strengths:**

1, This paper is well-motivated. In practical applications, not all data on clients are labeled, and how to leverage unlabeled data is important for FL.
2, The writing is clear and easy to follow.
3, The paper performs extensive experiments on three diverse datasets, encompassing varying scales, complexities, and domains. Moreover, the proposed method is compared against multiple baselines as well as state-of-the-art techniques. The experimental results consistently demonstrate improvements across different metrics, object categories, weather conditions, and data distributions. This comprehensive evaluation reinforces the effectiveness and robustness of the proposed method, highlighting its superiority over existing approaches in various scenarios.


**Weaknesses:**

1, This paper lacks discussion on the communication efficiency and scalability aspects of the proposed method, which are important considerations for its practical implementation. Specifically, it does not address the communication overhead associated with uploading only the backbone parameters or utilizing local exponential moving average (EMA) models. Furthermore, the paper does not investigate the performance of the method as the number of clients or the size of unlabeled data increases.
2, It could be better to compare or relate the proposed method with existing works on semi-supervised or self-supervised FL [1,2,3,4]. What are the advantages of the proposed method over these existing methods?
[1] Zhuang, Weiming, Yonggang Wen, and Shuai Zhang. "Divergence-aware federated self-supervised learning." arXiv preprint arXiv:2204.04385 (2022).
[2] Zhang, Fengda, et al. "Federated unsupervised representation learning." arXiv preprint arXiv:2010.08982 (2020).
[3] Wu, Yawen, et al. "Federated contrastive learning for volumetric medical image segmentation." International Conference on Medical Image Computing and Computer-Assisted Intervention. Springer, Cham, 2021.
[4] Dong, Nanqing, and Irina Voiculescu. "Federated contrastive learning for decentralized unlabeled medical images." International Conference on Medical Image Computing and Computer-Assisted Intervention. Springer, Cham, 2021.


**Questions:**

Please refer to the Weakness.

**Limitations:**

Please refer to the Weakness.

---

> ### Author Rebuttal · Authors · 2023-08-10
>
> ### W1. Communication efficiency and scalability aspect + # of clients and size of unlabeled data increases
> - We thank the reviewer for this thoughtful comment. Communication efficiency and scalability are indeed crucial considerations for any FL implementation. In our work, we are mindful of these aspects, and we have attempted to address them in the following ways:
>
> 1. **Communication Efficiency**: Our method communicates only the backbone parameters rather than the entire model, substantially reducing communication payload. As such, our selective communication. As the backbone typically constitutes a fraction of the model parameters, selective communication indeed reduces overhead. We have uploaded the relevant costs **in the attached pdf of the general response (Table D)**.
> 2. **Local EMA models**: The utilization of local EMA models helps mitigate communication constraints as it enables each client to independently generate pseudo labels. This reduces the need for frequent server updates, further improving communication efficiency.
> 3. **Scalability**: We appreciate your insight, and we agree that it is crucial to test our framework in a setting with a larger number of clients. To support this, we have conducted experiments with 1 server and 100 clients, as detailed in the supplementary material (**Appendix Section M**). The results presented in the appendix were obtained with a client sampling ratio of 0.1. Additionally, we put more results in **Table C in the attached pdf**.
>  - We would like to stress that our experiments are not merely theoretical or synthetic. They are conducted in a real-world setting, with genuine network communications occurring between 100 AWS virtual machines. We do this to underscore the practicality and applicability of our findings, and to convince the community that our proposed approach to personalized, semi-supervised FL is viable.
> - As for detailed discussions and evaluations regarding these aspects, due to space limitations in the main paper, we have allocated this content to the **Appendix**. But, if extra pages are granted for the camera-ready version, we will certainly incorporate more details about these practical aspects in the main text.
> - We hope this clarifies the reviewer's concern, and we appreciate the suggestion to focus more on these practical aspects. It will undoubtedly make our work more robust and closer to real-world applicability.
>
> ### W2. What are the advantages of the proposed method?
> - We appreciate the reviewer's suggestion to compare and relate our method with existing works on semi-supervised or self-supervised FL. Indeed, these are relevant and important works in the domain of federated learning. Here is how our method differentiates and improves over these works:
> 1. **Domain-Specificity**: Our work primarily targets object detection tasks in the FL setup, specifically focusing on the challenges presented by real-world scenarios such as autonomous driving. While the works suggested primarily concentrate on image classification or medical image segmentation, we are operating in a different problem domain with its unique challenges.
> 2. **Methodological Advances**: Our SSFOD (Semi-Supervised Federated Object Detection) framework proposes a novel federated semi-supervised learning approach tailored to object detection tasks. It integrates techniques like Personalized Pseudo Labeling, Orthogonal Enhancement, and Selective Communication to effectively leverage unlabeled data, enhance feature representation, and reduce communication overhead, respectively.
> 3. **End-to-End Solution**: Our work presents an end-to-end solution: problem formulation, benchmark setting, improved methods, and experimental results. We first formalize the problem, set up a benchmark for semi-supervised federated object detection, and finally propose a novel method to solve it.
>
> - That said, the suggested works do offer valuable insights and techniques in their respective fields. Some of their methodologies might be applicable and potentially beneficial to our work. For instance, Contrastive Learning techniques from [3] and [4] could be incorporated into our framework to further improve the feature representation capabilities of our model.
>
> - We appreciate the reviewer pointing out these references, and **we will certainly consider adding a discussion regarding these works in our paper**, focusing on how they relate to our work and how their methodologies could potentially be integrated into our framework. This will also serve to highlight the unique contributions of our work. We believe such comparisons would enrich the context of our paper and make it more comprehensive.

---

> > ### Author Response · Authors · 2023-08-16
> >
> > Dear Reviewer,
> >
> > - In response to the feedback, we've conducted rigorous additional experiments to enhance the depth and robustness of our work. This includes:
> > 1. Real-world experiments using 100 clients through 100 virtual machines.
> > 2. An extensive analysis of communication costs.
> > 3. Further ablation studies for a more comprehensive understanding.
> > - Additionally, we'd like to highlight that several points brought up during the review process have been addressed in our appendix. We've emphasized these in our updated responses for your convenience.
> > - Given the tight timeline, with the discussion phase concluding on Aug 21st at 1pm EDT, we kindly request you to review our responses. We believe our detailed responses provide clarity on the concerns raised. Your feedback is pivotal to the quality of our work, and we earnestly await your thoughts, especially since we have less than 6 days remaining.
> >
> > Thank you for your time and understanding.

---

### Official Review · Reviewer_8JqK · 2023-07-06

**Soundness:** 3 good
**Presentation:** 3 good
**Contribution:** 3 good
**Rating:** 5
**Confidence:** 5

**Summary:**

This work focuses on a practical application of federated learning, federated semi-supervised learning for object detection. It assumes that the server has labeled data and the clients only have unlabeled data. The proposed method is two-fold: selective training and orthogonal enhancement.

**Strengths:**

- This paper integrates multiple existing techniques for federated semi-supervised learning. It is technically sound.
- The paper is generally well-written, clear, and easy to follow.
- Experiments on three object detection datasets demonstrate the effectiveness of the proposed method.

**Weaknesses:**

- The novelty is somewhat limited. The novelty lies in the integration of existing methods and making them work on the target use case, while the key algorithms are more or less adopted from existing works.
- The evaluation scale can be extended to a larger number of clients. The majority of the experiments are run with 3 clients and only one experiment is run with 20 clients. The autonomous driving use case is more like cross-device FL. It’s important to evaluate on more clients with client sampling.
- Some existing works with related techniques are not discussed: [1] and [2] fixed the head and only trains the backbone like the selective training. [3] and [4] also uses EMA in local training.
- Since the first stage is to learn representation, a comparison could be done with federated self-supervised learning methods [3][4] for learning visual representations.
- The organization of section 3 is not very straightforward. For example, 'Personalized Pseudo Labeling for Unlabeled Clients' is more suitable in Section 4 instead of problem statement.

[1] Fedbabu: Towards enhanced representation for federated image classification, ICLR’22.

[2] Spherefed: Hyperspherical federated learning. ECCV’22

[3] Collaborative unsupervised visual representation learning from decentralized data. ICCV’21

[4] Divergence-aware federated self-supervised learning. ICLR’22.

**Questions:**

- Line 3 of the Algorithm conducts client sampling, while it seems that the experiments do not conduct client sampling.
- The server and client are assumed to be from the dataset. What would be the impact if the dataset in the server and clients are not from the same dataset? It would be more practical as we are unable to assume that we can collect the data in a server similar to the clients.

**Limitations:**

Limitations are not discussed in the paper. It could contains some of the aspects such as the scale of the experiments.

---

> ### Author Rebuttal · Authors · 2023-08-10
>
> Thank you for your comments. We address each below.
> ### W1. Novelty
> - We would like to respectfully disagree with the assertion that the novelty of our work is limited. When trying to naively apply existing techniques to the SSFOD setting, we observed notably poor performance, which led to our novel formulations. As we have pointed out in our paper, related FL research tends to focus on image classification and small image sizes, like in CIFAR. In contrast, our work tackles object detection, which is a more complex problem with larger and noisier images.
> - To elaborate on the novelty in the general response, we propose the first SSFOD framework, where the server has labeled data while clients only have unlabeled data. To the best of our knowledge, this is a novel problem formulation that has not yet been addressed in the literature. Rather than simply predicting classes in image classification, pseudo labeling in object detection involves predicting ground-truth boxes, applying non-maximum suppression, and assigning class labels to the predicted boxes. Because each step is intricate, constructing and training a useful pseudo labeler—especially in the presence of data heterogeneity—is far from simple (theoretical and ablation analysis is provided in the **Appendix**).
> - In the centralized learning setting, semi-supervised object detection usually refers to labeled data to correct the training of unlabeled data and perform domain adaptation with every batch. However, in our federated learning scenario, the clients are assumed to have no access to labeled data. Simply applying existing techniques in this context would result in suboptimal and unstable performance (**Alternate Training part in Table A** of the attached pdf & **Table 1 in the main paper**). We believe FedSTO is a novel contribution and hope that this explanation will reinforce this viewpoint.
>
> ### W2. Evaluation on a larger number of clients, client sampling & Q1. Client sampling
> - We agree that it is crucial to test our framework in a setting with a larger number of clients. To support this, we have conducted experiments with 1 server and 100 clients, as detailed in **the Appendix Section M & Table C in the attached pdf**.
> - We would like to stress that our experiments are not merely theoretical or synthetic. They are conducted in a real-world setting, with genuine network communications occurring between 100 AWS virtual machines. We do this to underscore the practicality and applicability of our findings, and to convince the community that our proposed approach to personalized, semi-supervised FL is viable.
>
> ### W3. Discussions for related techniques
> - Thank you for highlighting related works [1]-[4]. Regarding the freezing in [1] and [2], we have evaluated similar methods, especially those in "Personalized Federated Learning with Feature Alignment and Classifier Collaboration, ICLR 2023.”, which emphasizes enhancing feature alignment through freezing the head. Though similar, they focus on image classification while our work addresses the unique challenges of object detection. We list more results and the discussions of the ablation study about these methods [1, 2] in **Table 2, Appendix Section I & Appendix Table 7**.
> - For the EMA in local training as in [3] and [4], we acknowledge the similarities in terms of leveraging EMA models. However, while [3] and [4] seem to be inclined towards using pseudo labels or representations directly from the EMA model in the image classification task, our method involves generating annotations after the non-maximum suppression step using the EMA model's predictions in the object detection task.
> - **Given acceptance, we'll incorporate the references you provided in our final version.** This will certainly strengthen the connections to related methods.
>
> ### W4. 1st stage for self-supervised learning
> - Thank you for pointing out the potential comparison with federated self-supervised learning methods [3], [4]. While our approach has similarities with conventional self-supervised methods, our main focus was to progressively develop SSFOD, given the nascent state of research in this topic. We recognize the value of the cited works; however, they primarily target image classification. Directly applying their techniques to federated object detection introduces unique challenges not present in their original context. We are nonetheless enthusiastic about exploring a federated self-supervised backbone in our future research, and appreciate your insightful suggestion.
>
> ### W5. Not Straightforward Organization of section 3
> - Thank you for your keen observation on the organization of Section 3. We concur that "Personalized Pseudo Labeling for Unlabeled Clients" plays a pivotal role in our problem formulation. Given its significance, especially when juxtaposed with "data heterogeneity", we believe its initial discussion is warranted in Section 3. Our proposal is to retain an introductory segment on personalization in Section 3 and defer the deeper, algorithmic details to Section 4. This restructuring aims to strike a balance, addressing the problem's essence while streamlining technical content.
>
> ### Q2. Server and Client have different dataset
> - Thank you for the insightful comment. We concur that the server and client data from differing datasets introduce a valuable layer of heterogeneity.
> - Our current focus revolves around addressing complexities like weather-induced feature distribution skew and label density heterogeneity, particularly in applications like autonomous driving, detailed in **Appendix Section E**.
> - While the scenario you highlight—which could encompass out-of-distribution data or introduce new classes—is not directly addressed in our work, we anticipate our methods, such as personalized pseudo labeling, might provide some adaptability. However, this is a rich area warranting further exploration. In the future work, we will explore these scenarios more comprehensively.

---

> > ### Author Response · Authors · 2023-08-16
> >
> > Dear Reviewer,
> >
> > - In response to the feedback, we've conducted rigorous additional experiments to enhance the depth and robustness of our work. This includes:
> > 1. Real-world experiments using 100 clients through 100 virtual machines.
> > 2. An extensive analysis of communication costs.
> > 3. Further ablation studies for a more comprehensive understanding.
> > - Additionally, we'd like to highlight that several points brought up during the review process have been addressed in our appendix. We've emphasized these in our updated responses for your convenience.
> > - Given the tight timeline, with the discussion phase concluding on Aug 21st at 1pm EDT, we kindly request you to review our responses. We believe our detailed responses provide clarity on the concerns raised. Your feedback is pivotal to the quality of our work, and we earnestly await your thoughts, especially since we have less than 6 days remaining.
> >
> > Thank you for your time and understanding.

---

> > ### Comment · Reviewer_8JqK · 2023-08-19
> >
> > Thank you for your response. Most of the concerns are well addressed with provided explanations and supplemented experiments, the reviewer would like to increase the score to 5 and suggest the authors incorporate these content in the final version.

---

> > > ### Author Response · Authors · 2023-08-20
> > >
> > > Dear Reviewer,
> > >
> > > We are truly grateful for your thorough evaluation and the subsequent score adjustment. We will certainly incorporate the provided feedback into the final version of our manuscript to ensure its quality and coherence. Your insightful comments have been pivotal in refining our work.
> > >
> > > Thank you once again for your time and valuable input.

---

> ### Comment · Area_Chair_YTNz · 2023-08-18
>
> Dear Reviewer 8JqK, I'd like to kindly ask you to read the rebuttal provided by the authors. Please respond and/or update your rating if necessary. Thank you.  -AC

---

### Official Review · Reviewer_qq85 · 2023-07-10

**Soundness:** 3 good
**Presentation:** 3 good
**Contribution:** 3 good
**Rating:** 5
**Confidence:** 3

**Summary:**

This paper explores Semi-Supervised Federated Object Detection (SSFOD), a pioneering framework for distributed data sources with limited high-quality labels and non-IID client data, particularly in applications like autonomous driving. The authors present a two-stage strategy, FedSTO, encompassing Selective Training followed by Orthogonally Enhanced full-parameter training, to address data shift while representing the first implementation of SSFOD for clients with 0% labeled non-IID data. The proposed approach includes selective refinement of the detector backbone to avert overfitting, orthogonality regularization to enhance representation divergence, and local EMA-driven pseudo label assignment to produce high-quality pseudo labels.

**Strengths:**

The paper provides helpful figures, explores the valuable direction of semi-supervised Federated Object Detection (SSFOD), and presents an effective two-stage strategy called FedSTO for addressing data shift in a distributed data source. The proposed approach achieves state-of-the-art results in multiple datasets. Additionally, the paper provides a clear problem statement that is easy to understand.

**Weaknesses:**

However, there are a few areas for improvement before the paper can be considered for publication. First, the references list is incomplete as some essential references are missing, such as "Federated learning with label distribution skew via logits calibration."

**Questions:**

see weaknesses

**Limitations:**

The authors did not discuss limitation and ethical issues in the main body.

---

> ### Author Rebuttal · Authors · 2023-08-10
>
> Thank you for your review. We address your comments below.
> ### W1. Additional references  (such as `Federated learning with label distribution skew via logits calibration')
> - Thank you for pointing out the omission in our references list and for suggesting the inclusion of "Federated learning with label distribution skew via logits calibration." We appreciate your recommendation and acknowledge the significance of this work in the domain of federated learning.
> - However, we would like to kindly note that while both our research and the suggested reference operate within the realm of federated learning, the specific challenges and methodologies they address are quite distinct. The cited paper delves into addressing label heterogeneity in federated image classification through logits calibration. In contrast, our work primarily targets federated object detection, placing particular emphasis on weather-conditioned heterogeneity. Moreover, the inherent complexity of label skewness in object detection – which encompasses annotation, objects per image, and class heterogeneity – sets it apart from image classification. The nuances of the problems and the intricate differences between the two contexts may not make the suggested reference directly applicable to our study.
> - Nevertheless, we recognize the value of drawing connections between related yet distinct works in federated learning. Should we be granted additional pages for the camera-ready version, we are fully committed to incorporating the recommended reference into **Section 2.1 "Federated Learning (FL): Challenges and Advances"**, highlighting its relevance and distinction from our approach.
> - Once again, thank you for your valuable feedback and suggestions. We appreciate the time and effort you have dedicated to reviewing our paper.
>
> ### Limitations and ethical issues are not discussed
> - Thank you for pointing out the lack of an explicit "limitations and negative social impact" section in the main body of the manuscript.
> - Due to space constraints, and in an effort to maintain the coherence of the main text, we have extensively discussed the potential limitations and negative social implications **in the Appendix**. We understand the importance of addressing such concerns in contemporary AI research and have made a dedicated effort to ensure that these topics are addressed in detail, albeit in the supplementary section.
> - We believe that it is essential for readers and practitioners to be aware of the possible pitfalls, limitations, and societal implications of the proposed methods. We hope that the provided discussion in the Appendix serves this purpose. In subsequent versions of this paper, we'll strive to make such sections more prominent by referencing it in the main text. Thank you for your understanding and feedback.

---

> > ### Author Response · Authors · 2023-08-16
> >
> > Dear Reviewer,
> >
> > - In response to the feedback, we've conducted rigorous additional experiments to enhance the depth and robustness of our work. This includes:
> > 1. Real-world experiments using 100 clients through 100 virtual machines.
> > 2. An extensive analysis of communication costs.
> > 3. Further ablation studies for a more comprehensive understanding.
> > - Additionally, we'd like to highlight that several points brought up during the review process have been addressed in our appendix. We've emphasized these in our updated responses for your convenience.
> > - Given the tight timeline, with the discussion phase concluding on Aug 21st at 1pm EDT, we kindly request you to review our responses. We believe our detailed responses provide clarity on the concerns raised. Your feedback is pivotal to the quality of our work, and we earnestly await your thoughts, especially since we have less than 6 days remaining.
> >
> > Thank you for your time and understanding.

---

> > ### Comment · Reviewer_qq85 · 2023-08-16
> > **Thank you for the rebuttal**
> >
> > Thank you for the additional details and explanations, which resolved some of my concerns. Nevertheless, the overall quality is still borderline, and my rating keeps unchanged.

---

> > > ### Author Response · Authors · 2023-08-17
> > >
> > > Thank you for recognizing the efforts we've made to address the concerns you raised.
> > >
> > > Given that we believe we've addressed the sole concern you mentioned, we're curious about any additional aspects of our manuscript that might still seem ambiguous or unclear to you. **In the rebuttal, the only weakness pointed out was related to the reference list**. Your feedback is invaluable to us, and if there are other areas that need further clarification or improvement, we'd be eager to know.
> > >
> > > Your insights and guidance are instrumental, and we would appreciate any additional feedback you might have.
> > >
> > > Thank you once again for your time and consideration.

---

### Official Review · Reviewer_2CR4 · 2023-07-25

**Soundness:** 3 good
**Presentation:** 3 good
**Contribution:** 3 good
**Rating:** 6
**Confidence:** 4

**Summary:**

This paper presents a Semi-Supervised Federated Object Detection (SSFOD) framework featuring a two-stage training strategy, FedSTO, designed to address the challenges of heterogeneous unlabeled data in federated learning. The proposed framework employs selective training and orthogonality regularization with personalized EMA-driven pseudo-labeling to facilitate robust and diverse feature learning, enhancing object detection performance across multiple weather conditions and data distributions. The empirical results provide evidence of the merits of FedSTO over existing federated and semi-supervised learning methodologies. Notably, despite non-IID clients having no labels, FedSTO achieves performance comparable to fully supervised centralized models.

**Strengths:**

•	The paper introduces a solution for SSFL for object detection with a one-stage detector in non-IID weather conditions.
•	The paper outlines a new approach to SSFOD, combining existing methods and adapting them to the federated learning setting.
•	The paper is well-structured, has clear headings and subheadings, and is easy to follow.

**Weaknesses:**

1. The inclusion of a federated setting with fully labeled data on at least one dataset would have provided a valuable comparison to the proposed approach. Also, while the comparison with the partially supervised baseline helps establish the lower bound of server pretraining, it is not a fair comparison due to the significant difference in training data volume.
2. Although the models are evaluated on their respective datasets, it would have been beneficial to assess their performance on a global test set as well. Context-specific evaluations are necessary, but examining the models' generalization can help avoid the issue of overly specialized clients, which is a concern in personalization.
3. An interesting analysis would have been to investigate how the performance gap of the proposed method changes with increasing amounts of data. The reported performance of the Fully Supervised Centralized Yolo-v5 Large in the paper is notably lower compared to other papers on full-scale datasets (e.g., YoloV5s achieves 77.2 in https://arxiv.org/pdf/2108.11250v7.pdf). This raises the question of whether the low performance and the closure in performance with the centralized setting is due to the limited training data. It would be valuable to compare the proposed approach against these scenarios with more data available or if possible employing a full server pretraining.
4. Regarding Cityscape, since the dataset does not provide precise weather information for each annotation, the data is distributed uniformly at random, and it's not clear how it is non-IID. Also, the non-IID aspect mentioned in the paper is limited to addressing the skew in feature distribution induced by weather variations. The authors could have utilized foggy-Cityscape and KITTI datasets o obtain a more realistic non-IID setting.
5. The proposed SSFOD problem setting is similar to unsupervised domain adaptative object detectors and test-time domain adaptation. Even, the ingredients of the proposed solution such as EMA are also known in the semi-supervised OD and domain adaptation literature.  To me, the only difference is that the model parameters are updated on local clients instead of central weight update.   Please clarify the differences and advantages of the proposed solution compared to similar domain-adapted object detection methods.
6. Many key components such as FPT with orthogonal regularization are adapted from existing literature such as [16].  Although I agree that the proposed problem setting is unique and novel, it is also important to clarify the novel technical contributions of the proposed solution.
7. What is the rationale behind using Yolov5?  It will be beneficial to show that the proposed solution is generalizable on other families of object detectors such as Faster-R CNN and recent transformer-based object detectors such as DETR/Deformable DETR.
8. mAP@0.5 is not an ideal evaluation metric to qualify localization capabilities. Please report mAP@0.75 and COCO-style mAP for a better understanding of the precise localization capabilities of the model.
9. EMA, pseudo-label assignment on unlabelled data needs to be better explained. What are the training loss functions used for backbone weight update during the selective training step?
10. It is mentioned that the proposed solution uses augmentations such as Mosaic, left-right flip, large-scale jittering. But, it is not clear how are the corresponding ground truth box positions determined in the case of unlabeled images.
11.  As the authors mention that SSFOD introduces additional computational overhead, it is desired to quantify the computational resources employed.
12. The paper contains some errors, such as a missing dot in Line 139 and the use of "IID" in Table 5.

**Questions:**

Please see the weaknesses mentioned above.

**Limitations:**

The paper does not have an explicit section for "limitations and negative social impact", but that is not a major concern for the research topic studied in the paper.

---

> ### Author Rebuttal · Authors · 2023-08-10
>
> Thank you for your valuable feedback in helping refine our work.
> ### W1. Results of fully labeled FL & W8. mAP@0.75
> - As you suggested, we added fully-supervised IID and non-IID FL in Table A of our Global response's PDF as well as comprehensive evaluations using mAP@0.75. These results demonstrate that our FedSTO approach achieves competitive performance even when it only has access to 25% of the labels.
> - The partially supervised baseline establishes lower bounds of server pretraining and shows utility of unlabeled data. We agree that the strength of FedSTO is highlighted by 3 other baselines — significant improvements against vanilla SSFL and close performance to fully-supervised centralized and FL results.
>
> ### W2. Global performance
> - We concur in analyzing the generalizability of our models to other data distributions. Table E of our Global response's PDF shows FedSTO's competitive global performance.
>
> ### W3. Lower performance compared to the previous work
> - Upon inspection, the Yolov5's 0.772 in your referenced paper (https://arxiv.org/pdf/2108.11250v7.pdf) is only for the Car class. As listed in Table 4 of our paper, our centralized model reaches 0.788 and FedSTO reaches 0.740 for the Car class. Thus, our paper's performance is comparable (especially since we only have access to 25% of the labels).
>
> ### W4. Non-IIDness on Cityscapes
> - Your observation is correct that Cityscapes was distributed uniformly at random (stated in  line 260 of our main text). As you suggested, we would like to incorporate experiments with Foggy Cityscapes, space permitting. As our paper pioneers the exploration of SSFOD, we chose to demonstrate the efficacy of our approach on standard settings of datasets and anticipate future extensions to more nuanced settings.
>
> ### W5. Differentiation from previous domain-adapted object detection (OD) methods
> - Although inspired by past unsupervised domain adaptation methods, there are several major differences in our work. A novel challenge we address is the separation of labeled and unlabeled data. Conventional domain-adapted OD methods use labeled data instances to conduct SSL. Our scenario, where each client only holds unlabeled data and cannot access labels, presents a unique and complex problem.
> - To overcome this, FedSTO uses personalized pseudo labelers and orthogonal enhancements:
>   - Go against conventionally used global models and demonstrate that local EMA models are stronger pseudo labelers.
>   - Stabilize training with only unlabeled data by using the alternate server and local pseudo labelers.
>   - Introduce warm-up and alternate training phases which yield personalization and generalization.
>   - Mitigate instability from training exclusively with unlabeled data by employing selective training.
> - Theoretical analysis is included in the Appendix.
>
> ### W6. Novel technical contribution
> - Although we build on prior research, their context and application differ significantly. A detailed theoretical explanation is in **Section F of our Appendix**. The greater challenge in using single-stage detectors, along with strategic loss application, signifies novel technical contributions.
> - For example, even though the penalizing loss may appear similar to existing literature such as [16] cited in our main paper, the intent behind its usage differs vastly. The focus of [16] was to enhance DNNs by applying across various layers of the architecture. In contrast, we have strategically applied the loss to the neck and head of our architecture with a specific aim: robustness of detection quality to heterogeneities among client datasets. Similarly, components like FPT with orthogonal regularization were primarily tested on image classification tasks, whereas we have adapted it to the more challenging domain of object detection.
>
> ### W7. Rationale behind Yolov5
> - Given our problem setting's novelty, we focused on single-stage detectors as they jointly achieve high performance, faster inference times, and small model sizes, which are crucial for real-world applications such as autonomous driving. Among single-stage detectors, Yolo models are a de facto standard. In addition, the nature of FL demands a focus on models that do not incur excessive communication costs. The suggested models, unlike Yolo, do not necessarily satisfy the fast inference or low communication cost criteria.
>
> ### W9. EMA pseudo label assignment + training loss
> - Thank you for your interest in the pseudo label mechanism and training loss. Due to space constraints, we provide explanations in **Appendix G and H**. As a brief summary, EMA constructs an enhanced local model that smoothens fluctuations during model updates. The EMA-based model serves as a local pseudo labeler to label unlabeled data held by each client. This labeling is essential for performing learning tasks on the unlabeled data. We employ the combination of objectiveness, bounding box, and classification losses included in the standardized Yolov5 loss. We train only on datapoints whose pseudo labels have a high confidence score.
>
> ### W10. Ground truth boxes for augmented unlabeled images
> - Ground truth box positions for unlabeled images are determined through predictions provided by our EMA pseudo labeler. They are subjected to data augmentation in parallel with image data. Thus, we generate augmented image label pairs from unlabeled images, which are then used for further training.
>
> ### W11. Computational resources
> - In our study, while SSFOD has inherent complexities due to pseudo labeling, it did not significantly increase memory use compared to fully-supervised FL; we utilized the same V100 GPU throughout. Furthermore, FedSTO reduces communication costs by 20.52% by freezing the model's neck, as seen in Table D of our response pdf. Going forward, we will aim for an optimal balance between performance and efficiency.
>
> ### Limitations
> - Due to space constraints, potential limitations and societal implications are in the Appendix.

---

> > ### Comment · Reviewer_2CR4 · 2023-08-15
> > **Thank you for the rebuttal and additional experiments.**
> >
> > Thanks for the well-organized rebuttal from the authors. The feedback solves most of my concerns regarding the paper, so I am tending to increase my rating to Weak Accept. But, I also would like to hear from other reviewers regarding their thoughts on the rebuttal and if they have some open issues.

---

> > > ### Author Response · Authors · 2023-08-15
> > >
> > > Dear Reviewer 2CR4
> > >
> > > Firstly, we'd like to express our sincere gratitude for taking the time to review our rebuttal in detail and for your consideration in adjusting your rating. Your initial feedback was instrumental in helping us clarify and enhance the aspects of our paper.
> > >
> > > We understand and respect your desire to gather collective insights from other reviewers. If there are further questions or any additional concerns post-discussions, please feel free to raise them. We're committed to addressing all aspects to ensure the clarity and quality of our work.
> > >
> > > Again, thank you for your constructive feedback, and we truly appreciate your open-minded approach to our rebuttal.

---

> > > > ### Author Response · Authors · 2023-08-20
> > > >
> > > > Dear Reviewer,
> > > >
> > > > Thank you for acknowledging our efforts in the rebuttal. Now that all reviewers have provided their feedback, may we kindly remind you to revisit the discussions and share any further thoughts or concerns you might have? Your insights are invaluable to the refinement of our paper.
> > > >
> > > > Warm regards

---

### Author Rebuttal · Authors · 2023-08-10

We extend our gratitude to all the reviewers for providing comprehensive and thoughtful feedback on our manuscript. We appreciate your valuable insight into the strengths and areas for improvement of our work.

### Summary of Strengths cited by Reviewers
- **Novelty in Approach and Framework:** We are encouraged by the observations of **Reviewers 2CR4** and **Reviewer qq85** noting the significant strength of our proposed SSFOD and FedSTO approaches, particularly in non-IID weather conditions. While it may seem prior methods tackled similar problems for image classification, these methods perform poorly in SSFOD's substantially more complex and practically relevant setting. The novel components of FedSTO — two-stage selective training and high-quality personalized pseudo-labelers — substantially boost performance compared to simply applying prior work to SSFOD (**we present ablation studies and theoretical analysis in Tables 1 & 2 of the main text and Sections F - J of the Appendix**). Indeed, FedSTO nears the performance of centralized learning models even with only having access to 25% of the labels and non-IID data.
- **Impact**: We appreciate **Reviewers AM8c, Reviewer 1XGa**, and **Reviewer qq85** for noting the important motivation and impact of our work. FedSTO reaches competitive performance while also exhibiting crucial real-world benefits of not sharing data to a central server and not requiring labeling at the edge. Preserving user privacy and reducing training cost are necessary improvements for feasibility in the real-world for a wider range of applications.
- **Clarity, Structure, and Presentation:** We are pleased to note the collective positive feedback on the clarity and presentation of our work, as acknowledged by **Reviewers 2CR4**, **Reviewer qq85**, **Reviewer 8JqK**, and **Reviewer AM8c**.
- **Technical Soundness and Integration:** **Reviewers 8JqK** and **Reviewer AM8c**'s emphasis on our paper's technical solidity is greatly appreciated. Presenting multiple techniques for semi-supervised FL and providing theoretical analysis (in the Appendix) while ensuring the coherence of the overarching framework was one of our primary objectives.
- **Extensive Experiments and Comparative Analysis:** We are encouraged by the feedback from **Reviewers AM8c** and **Reviewer 1XGa** regarding our empirical approach. We conducted comprehensive experiments across diverse datasets and made comparisons against multiple baselines—as well as SOTA techniques—to showcase the efficacy and robustness of FedSTO. **Reviewer 1XGa** specifically acknowledges the effectiveness of selective training in establishing a robust backbone for object detectors in FL, particularly its ability to reduce biases and heterogeneities.
- **Relevance to Practical Applications:** The observations made by **Reviewer AM8c** highlight our work's significance in real-world scenarios, where not all client data may be labeled. Leveraging unlabeled data in FL remains an imperative challenge, and our work attempts to provide tangible solutions in this direction. FedSTO also reduces communication cost by 20.52% in comparison to conventional FL algorithms, as **detailed in the attached PDF of tables**.

### Core Contributions of Our Work
- **Semi-Supervised Federated Object Detection (SSFOD)**: We are glad that **Reviewers 2CR4**, **Reviewer qq85**, **Reviewer 1XGa**, and **Reviewer AM8c** acknowledged the novelty of SSFOD, especially the challenging landscape of FL for distributed data sources with limited labeled data and non-IID client data. Our approach is positioned to significantly benefit applications such as autonomous driving. Not requiring labels in non-IID settings promotes not only cost reductions in training but also importantly preserves privacy as data is retained at the edge.
- **FedSTO**: As pointed out by **Reviewers 2CR4**, **Reviewer qq85**, and **Reviewer 1XGa**, the hallmark of our paper is the introduction of the two-stage training strategy, FedSTO, specifically tailored for clients with 0% labeled non-IID data. This strategy, with its selective training and orthogonally enhanced full-parameter training, is designed to tackle data shifts and represents a pioneering implementation of SSFOD.
- **Selective Training & Orthogonal Enhancement**: **Reviewers AM8c** and **Reviewer 8JqK** have emphasized our distinctive approach of selective training, aimed at refining the detector backbone to prevent overfitting, and orthogonal enhancement, which fosters representation diversity and robustness. These strategies collectively enable our framework to improve generalization capabilities across diverse data distributions.
- **Personalized Pseudo-labeling through EMA**: Both **Reviewers qq85** and **Reviewer AM8c** brought attention to our personalized EMA-driven pseudo-label assignment, a novel contribution that ensures the generation of high-quality pseudo labels for object detection tasks. This component elevates the performance of object detectors in a semi-supervised, non-IID FL context.
- **Empirical Validation and Superior Performance:** As recognized by **Reviewer 2CR4**, **Reviewer AM8c**, and **Reviewer 1XGa**, the empirical validation of our method across multiple datasets (BDD100K, Cityscapes, and SODA10M) demonstrates the superiority of our proposed methodology over existing federated and semi-supervised learning methods. The notable improvements in mAP@0.5 metrics when compared to baseline models further accentuates the efficacy of our approach.

### Things in Pdf
In the attached PDF, we included the following results:
- Results for fully supervised FL
- mAP@0.75 results
- Global model's performance
- Results with 100 clients and various sampling ratio

In light of the feedback, we are committed to refining our manuscript further, addressing any lingering queries, and incorporating your valuable insights into the final version of our paper.

---

### Decision · Program_Chairs · 2023-09-21

**Decision:**

Accept (poster)

**Comment:**

The reviewers are positive on the technical contributions and impact of the proposed federated learning framework for object detection. During the review process, issues like novelty and proper comparisons/evaluation were raised, which were later addressed by the authors during the rebuttal. Since all reviewers lean toward the acceptance of this paper, I think it is above the threshold for publication.